# Evidence for opposing selective forces operating on human-specific duplicated *TCAF* genes in Neanderthals and humans

PingHsun Hsieh [1✉], Vy Dang[1,4], Mitchell R. Vollger [1], Yafei Mao [1], Tzu-Hsueh Huang[1], Philip C. Dishuck [1], Carl Baker[1], Stuart Cantsilieris[1,5], Alexandra P. Lewis[1], Katherine M. Munson [1], Melanie Sorensen[1], AnneMarie E. Welch[1,6], Jason G. Underwood[1,2] & Evan E. Eichler [1,3✉]

TRP channel-associated factor 1/2 (TCAF1/TCAF2) proteins antagonistically regulate the cold-sensor protein TRPM8 in multiple human tissues. Understanding their significance has been complicated given the locus spans a gap-ridden region with complex segmental duplications in GRCh38. Using long-read sequencing, we sequence-resolve the locus, annotate full-length *TCAF* models in primate genomes, and show substantial human-specific *TCAF* copy number variation. We identify two human super haplogroups, H4 and H5, and establish that *TCAF* duplications originated ~1.7 million years ago but diversified only in *Homo sapiens* by recurrent structural mutations. Conversely, in all archaic-hominin samples the fixation for a specific H4 haplotype without duplication is likely due to positive selection. Here, our results of *TCAF* copy number expansion, selection signals in hominins, and differential *TCAF2* expression between haplogroups and high *TCAF2* and *TRPM8* expression in liver and prostate in modern-day humans imply *TCAF* diversification among hominins potentially in response to cold or dietary adaptations.

[1] Department of Genome Sciences, University of Washington School of Medicine, Seattle, WA, USA. [2] Pacific Biosciences (PacBio) of California, Incorporated, Menlo Park, CA, USA. [3] Howard Hughes Medical Institute, University of Washington, Seattle, WA, USA. [4]Present address: Institute for Cell and Molecular Biology, University of Texas, Austin, TX, USA. [5]Present address: Centre for Eye Research Australia, Department of Surgery (Ophthalmology), University of Melbourne, Royal Victorian Eye and Ear Hospital, East Melbourne, VIC, Australia. [6]Present address: The Walter and Eliza Hall Institute of Medical Research, Melbourne, VIC, Australia. ✉email: hsiehph@uw.edu; eee@gs.washington.edu

Gene duplication contributes significantly to molecular evolution by providing the raw material for genetic novelty and organismal adaptation[1,2]. In the human lineage, recently duplicated regions corresponding to segmental duplications (SDs) are known to give rise to new genes contributing to synaptogenesis, neuronal migration, and neocortical expansion[3–7] that distinguish human from other ape species. Although relatively few in number, some of the largest genetic changes that differentiate humans from archaic hominins involve gene-rich SDs[8,9]. Among human-specific SD genes, the duplications of *TCAF1*/*TCAF2* are particularly intriguing. These genes encode TRP channel-associated factors that bind to the protein TRPM8 (transient receptor potential cation channel subfamily M member 8)—the ion channel acting as thermal sensor in peripheral somatosensory neurons[10] and are thought to be under positive selection in Eurasian populations[11]. Both TCAF1 and TCAF2 proteins interact directly with the TRPM8 channel but have antagonistic effects in TRPM8 gating and trafficking into the plasma membrane: while TCAF1 facilitates the TRPM8 channel opening and migration, the activity of TRPM8 is completely suppressed by TCAF2[12]. These results suggest that the relative abundance of TCAF proteins and their competition in binding TRPM8 is likely critical in the physiological regulation of the channel activity[12].

The *TCAF* family originated from an ancient gene duplication event at the basal of mammalian phylogeny and remained single-copy genes throughout much of their evolution[8,12]. Within the human lineage, subsequent duplications over the last few million years have changed the copy number of both *TCAF1* and *TCAF2*. In the human reference genome GRCh38, *TCAF1* and *TCAF2* are embedded and span within a complex region of large, highly identical SDs (>99.5%) consisting of >250 thousand base pairs (kbp) in sequence and an annotated gap at chromosome 7q35. Studying these SDs is particularly problematic because of the high sequence identity and missing sequences in reference genomes[13–15]. Using large-insert bacterial artificial chromosome (BAC) clones from a haploid hydatidiform mole cell line, we recently assembled a sequence-resolved haplotype spanning over this region and characterized three sets of SDs present in both direct and inverted orientations. Copy number estimations from read-depth analyses suggest that *TCAF* duplications are completely missing in archaic hominins, such as Neanderthal and Denisovan, and nonhuman great apes[8]. In addition, while their sequence analysis suggested that a single full-length *TCAF1* and *TCAF2* exist at the locus, respectively, additional *TCAF1*/*TCAF2* copies appear to be truncated or incomplete[8].

In the present study, we systematically explore the haplotype structure of the *TCAF* locus in order to study its diversity, annotate the genes, and infer its evolutionary history in the context of selection. We generate 15 sequence-resolved haplotypes from both human and nonhuman primate BAC libraries as well as full-length non-chimeric (FLNC) transcript data from seven tissues using the Pacific Biosciences (PacBio) single-molecule, real-time (SMRT) long-read sequencing technology. We leverage a large collection of over 1,100 publicly available high-coverage Illumina genomes from modern, and archaic humans as well as nonhuman primate samples and document the global diversity of *TCAF* SDs. Finally, we integrate both phylogenetic and population genetic inference approaches to reconstruct the evolution of the *TCAF* SDs in primates and identify putative signals of balancing selection in Native American, Melanesian, and Siberian populations as well as possible positive selection in both Neanderthal and Denisovan. We provide evidence for differential expressed quantitative trait loci (eQTLs) between two *TCAF* super haplogroups in liver, prostate, thyroid, and tibial nerve tissues. This study is one of the detailed genetic investigations of human-specific SDs shedding potential new insights into structural adaptations important in thermal regulation.

## Results

**Diversity of *TCAF* copy number in human and nonhuman primates.** We applied a read-depth-based genotyper[16] to reevaluate copy number diversity of *TCAF* SDs (duplication segments A [DupA], B [DupB], and C [DupC]; Fig. 1a) in a collection of recently published diverse human and nonhuman samples[17–23]. The set includes high-coverage Illumina genomes from a global panel of 1,102 human samples, four archaic hominins, and 71 nonhuman great apes. Consistent with the previous studies[8], all archaic hominin and nonhuman primate samples are fixed for diploid copy number (CN) 2 (Fig. 1b). Among modern human samples, the overall diploid copy number estimates of the three *TCAF* SDs range from two to eight copies and are highly correlated among each other (Supplementary Fig. 1), indicating that these three SDs likely appear as a cassette in the evolution of this locus. In contrast to archaic humans where there is a prediction of a single copy of this locus, we estimate that ~98% of modern humans carry more than two copies (Fig. 1b and Supplementary Fig. 1), suggesting that most of the diversity emerged during the divergence of hominins.

Among humans, we observed a wide range of variations in the *TCAF* copy number across different geographic locations (Fig. 1b, c). While the median copy numbers are consistently CN4 (with the exception of Native Americans [median = 3]), African samples on average carry more copies (mean CN = 4.6) than other populations (Fig. 1b). We tested for copy number stratification by applying both the $V_{ST}$ statistic[24] and copy number differentiation test[25]. While there is little evidence for population differentiation among pairs of super populations, a reduction in diploid copy number among Native American samples distinguishes this group from all others ($V_{ST} > 0.21$, Bonferroni $p$ value for the copy number differentiation test <0.0014; Supplementary Data 1).

The observation that all diploid CN2 samples (i.e., without duplications) are present solely among non-African populations raises the question as to whether it is due to archaic introgression. To address this, we compared single-nucleotide variation at 20 kbp unique diploid sequences flanking the *TCAF* SD locus (chr7:143,501,000–143,521,000, chr7:143,875,000–143,895,000 [GRCh38]) between archaic and modern human genomes. In general, the archaic flanking haplotypes are similar to those present in modern human samples, including the AFR samples (Supplementary Fig. 2). Focusing on sites that are fixed in derived alleles in either the Neanderthal or Denisovan samples, the numbers of derived allele counts in the CN2 samples are comparable to those drawn randomly from African samples ($p = 0.276$; 1000 non-parametric bootstrap samples). These results suggest that non-African CN2 haplotypes unlikely arose as a result of introgression from Neanderthal or Denisovan.

**Characterization of *TCAF* haplotype diversity.** These near-perfect duplications have hampered not only genome assembly but also the characterization of underlying genetic diversity as well as our understanding of the *TCAF* evolution. An investigation of four recently published long-read whole-genome assemblies, for example, showed that all four failed to create contiguous sequences for both haplotypes spanning over the *TCAF* SDs (Supplementary Fig. 3 and Supplementary Data 2). To resolve this, we selected large-insert BAC clones and used PacBio long-read sequencing to generate 15 high-quality sequence-resolved haplotypes from eight human samples and three nonhuman

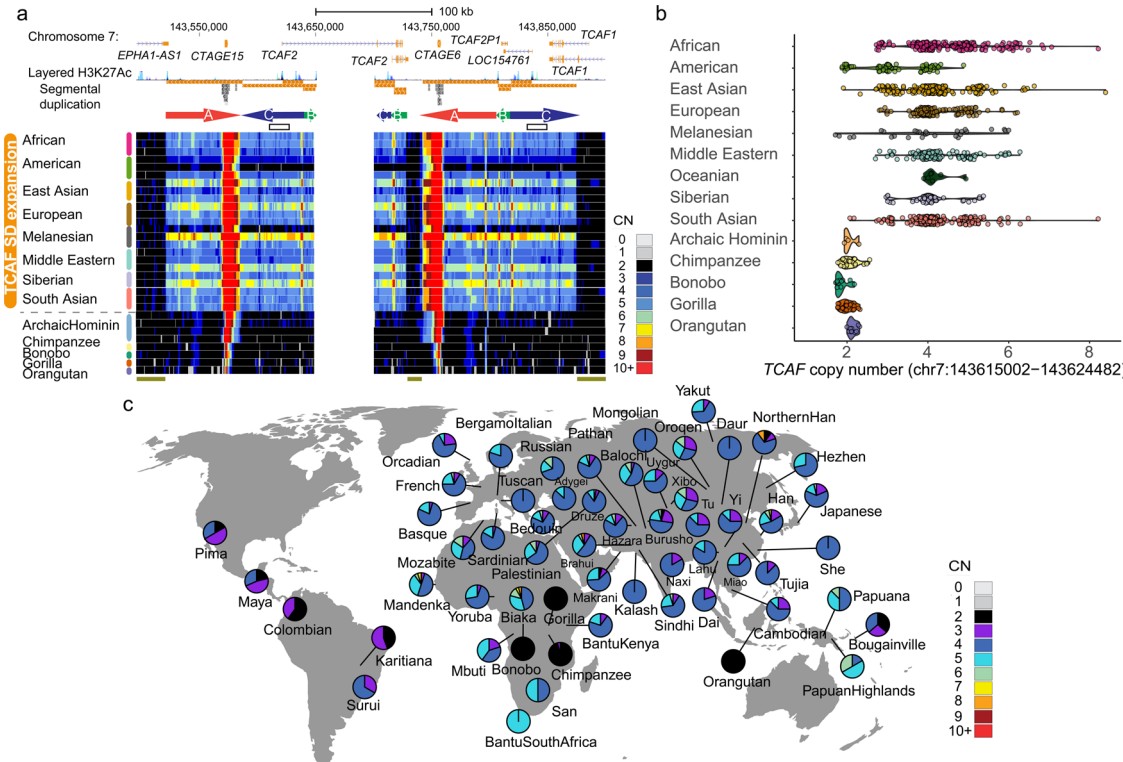

**Fig. 1 Copy number variation of *TCAF* SDs in a collection of diverse human and nonhuman samples.** Copy numbers were estimated using a read-depth-based genotyping method. **a** Copy number heatmaps, where each row represents the CN of a sample over the *TCAF* locus. The colored arrows (A, B, and C) represent the three major SD blocks in this region. The white area in the middle represents the gap present in the human reference genome (GRCh38). The two white boxes show a putative interlocus gene conversion event that correlates with latitudinal locations of populations (Supplementary Figs. 16–20). Dark green bars at the bottom indicate the unique diploid sequences used for downstream phylogenetic and population genetic analyses (chr7:143,501,000–143,521,000, chr7:143,729,525–143,741,525, chr7:143,875,000–143,895,000 [GRCh38]). **b** Distributions of the overall *TCAF* CN genotypes among samples from nonhuman great apes, archaic hominins, and modern humans using a representative region (chr7:143,615,002–143,624,482). **c** Geographic distribution of the overall *TCAF* CN genotypes in the 54 Human Genome Diversity Project (HGDP) populations and the nonhuman great ape samples. Geo-coordinates for the populations are listed in Supplementary Data 8. Pie charts show the CN frequency distribution for a given population. The map was created using the R packages rgdal (v1.5), scatterpie (v0.1.6), and Natural Earth. Note that the color scheme is slightly different from those in the CN heatmaps in Fig. 1a.

**Table 1 Summary of 15 assembled *TCAF* haplotypes constructed using large-insert BAC libraries and long-read sequencing.**

| Haplotype ID | BAC library (species or population) | Length (bp) | Length of *TCAF* SD cassettes (bp) | Copy number of *TCAF* SD cassettes | % GC | Haplogroup |
|---|---|---|---|---|---|---|
| CHM1 | CHM1 | 368,013 | 277,806 | 2 | 39.83 | Haplogroup 2-1 |
| VMRC53_hapA | NA12878 (European) | 433,048 | 277,806 | 2 | 39.56 | Haplogroup 2-2 |
| VMRC53_hapB | NA12878 (European) | 425,306 | 273,483 | 2 | 39.63 | Haplogroup 3-2 |
| VMRC61_hapA | HG00732 (Puerto Rican) | 337,690 | 277,856 | 2 | 39.91 | Haplogroup 2-2 |
| VMRC61_hapB | HG00732 (Puerto Rican) | 435,583 | 405,366 | 3 | 39.71 | Haplogroup 4 |
| VMRC62_hapA | HG00733 (Puerto Rican) | 395,405 | 277,854 | 2 | 39.60 | Haplogroup 2-2 |
| VMRC64_hapA | NA19240 (Yoruba) | 323,367 | 260,853 | 2 | 39.94 | Haplogroup 2-2 |
| VMRC64_hapB | NA19240 (Yoruba) | 348,654 | 277,808 | 2 | 40.07 | Haplogroup 2-2 |
| VMRC66_hapA | NA19434 (Luhya) | 496,357 | 406,131 | 3 | 40.00 | Haplogroup 5 |
| VMRC69_hapA | HG00514 (Han Chinese) | 387,079 | 277,712 | 2 | 39.29 | Haplogroup 3-1 |
| VMRC73_hapA | GM10539 (Melanesian) | 247,628 | 145,427 | 1 | 39.93 | Haplogroup 1 |
| VMRC73_hapB | GM10539 (Melanesian) | 222,558 | 145,424 | 1 | 39.85 | Haplogroup 1 |
| CH251_contig | CH251 (Pan troglodytes) | 273,442 | 127,988 | 1 | 39.90 | Ancestral-like |
| CH277_contig | CH277 (Gorilla gorilla) | 241,956 | 140,234 | 1 | 39.98 | Ancestral-like |
| CH250_contig | CH250 (Rhesus macaque) | 225,909 | 140,184 | 1 | 40.49 | Ancestral-like |

BAC clones were selected and sequenced using the PacBio long-read sequencing technology and assembled into individual haplotypes (Methods). Copy number of *TCAF* segmental duplication (SD) cassettes and the classification for individual haplotypes were determined by Miropeats and sequence alignment analysis (Fig. 2 and Supplementary Figs. 5–13).

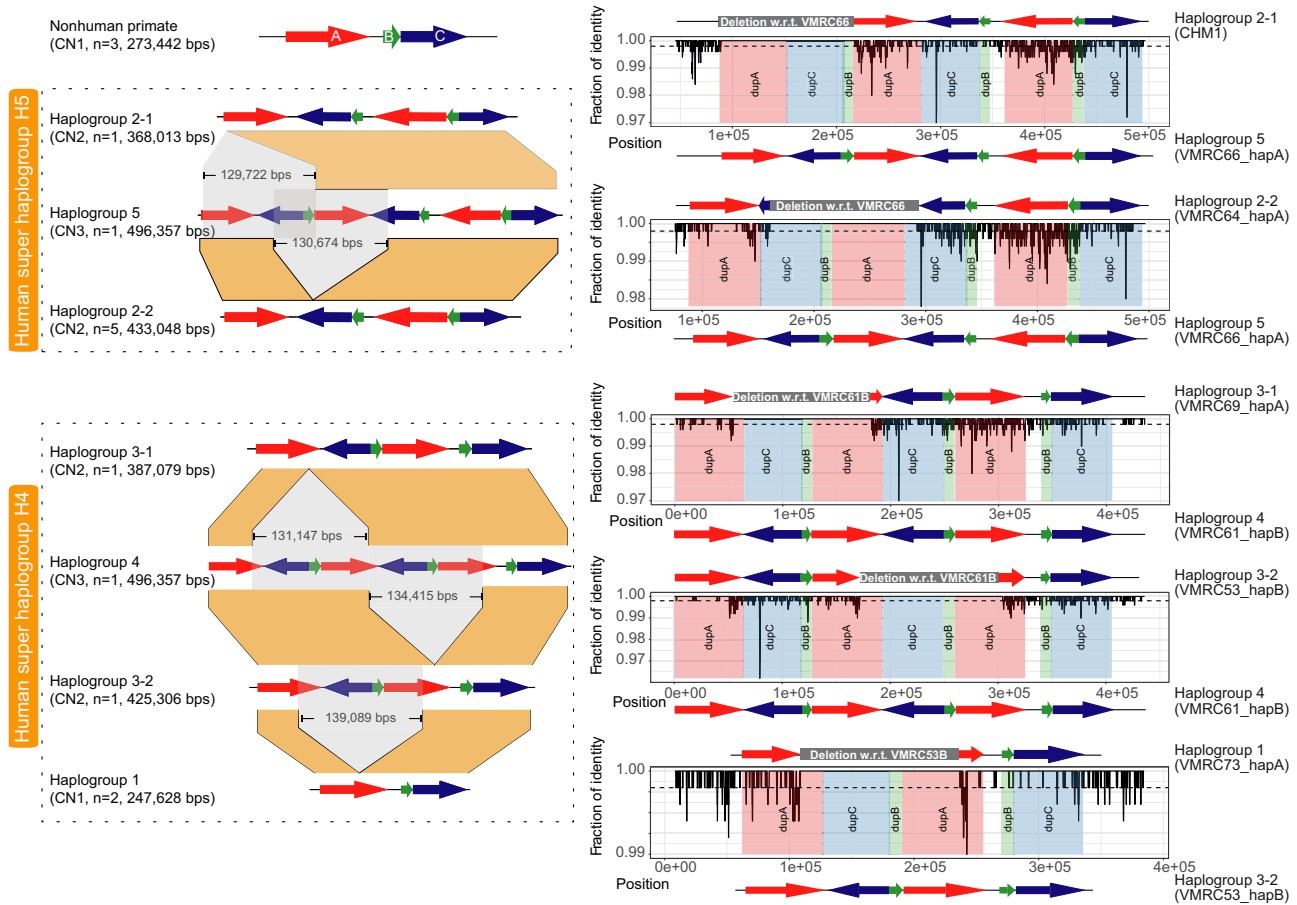

**Fig. 2 Complex structural haplotype diversity at the 7q35 *TCAF* locus in humans.** Left panel: Schematic representations of the major- (super haplogroups H4 and H5) and sub-haplogroups identified from 15 targeted BAC long-read assemblies. Colored arrows are *TCAF* SDs, and the gray and orange areas indicate putative deletion events and orthologous sequences between haplogroups. For example, despite the similarity in structure, Haplogroup 2-1 and Haplogroup 2-2 likely resulted from different deletion events relative to Haplogroup 5 based on comparative sequence data. Right panel: The comparative sequence analysis between individual haplogroups shows pairwise sequence identities (black lines) over 500 bp windows (sliding by 100 bp). With each plot, colored rectangles correspond to *TCAF* SDs, while the white region embedded between DupA (red) and DupB (green) blocks represents a 12.3 kbp single-copy unique sequence. Gray rectangles above each sequence identity plot indicate the locations of putative structure variants (Supplementary Data 3).

primates, including chimpanzee, gorilla, as well as an Old World rhesus macaque monkey (Table 1, Methods).

A sequence comparison analysis among these haplotypes confirms the complex organization of these *TCAF* SDs[8] and further reveals considerable structural diversity in humans when compared to nonhuman primates (Fig. 2 and Supplementary Figs. 4–12). While all the nonhuman primate haplotypes carry a single copy for the *TCAF* SD cassette and single full-length *TCAF1* and *TCAF2* genes (Supplementary Figs. 4, 11–12), among the 12 human haplotypes, we identify five distinct haplogroups that carry one to three copies for the SD cassette, which range from 145–406 kbp in length (Fig. 2, Table 1). The two largest human haplogroups (Haplogroups 4 and 5) both carry three copies of the SD cassette (Fig. 3 and Supplementary Fig. 5) but differ by a 100 kbp inversion (Haplogroup 5:337,305–437,635; Supplementary Fig. 5). The Haplogroup 5 organization is mostly consistent with the current human reference genome (GRCh38), which has a gap in the middle of expanded complex SDs in the reference (Fig. 3). The sequence of Haplogroup 5, thus, eliminates the gap by adding 103,616 bp (Haplogroup 5:216,541–320,157) in the human reference.

Most of the sequence-resolved haplotypes (8 out of 15) consist of two copies of the *TCAF* SD cassette (Haplogroups 2 and 3;

Fig. 2 and Supplementary Figs. 6–9). These two haplogroups can be structurally distinguished by a similar 100 kbp inversion as identified between Haplogroups 4 and 5. We, therefore, use this inversion polymorphism to classify two super haplogroups (H4 and H5, Fig. 2). The shortest human BAC haplotypes (Haplogroup 1; Fig. 2 and Supplementary Fig. 10) that we assembled consist of a single copy of the SD cassette and are similar in structure to those found in nonhuman primates. To further refine the relationship among these assembled haplotypes, we performed a series of sequence alignment analyses among different pairs of haplotypes (Fig. 2). Based on the organization of the *TCAF* SDs, we conservatively determined that Haplogroups 2 and 5 are closely related and so are Haplogroups 3 and 4. Among Haplogroup 2, two subgroups, Haplogroup 2-1 ($n = 1$) and Haplogroup 2-2 ($n = 5$), were further separated and likely emerged due to two independent events, where each involves a different 130 kbp deletion, with respect to Haplogroup 5 (Fig. 2 and Supplementary Data 3). Similarly, Haplogroup 3 can also be further classified into two subgroups, Haplogroup 3-1 and Haplogroup 3-2, because we predict that they likely derived from two different large deletions with respect to Haplogroup 4 (Fig. 2 and Supplementary Data 3). Finally, while Haplogroup 1 has a similar structure to the nonhuman primate haplotypes, we

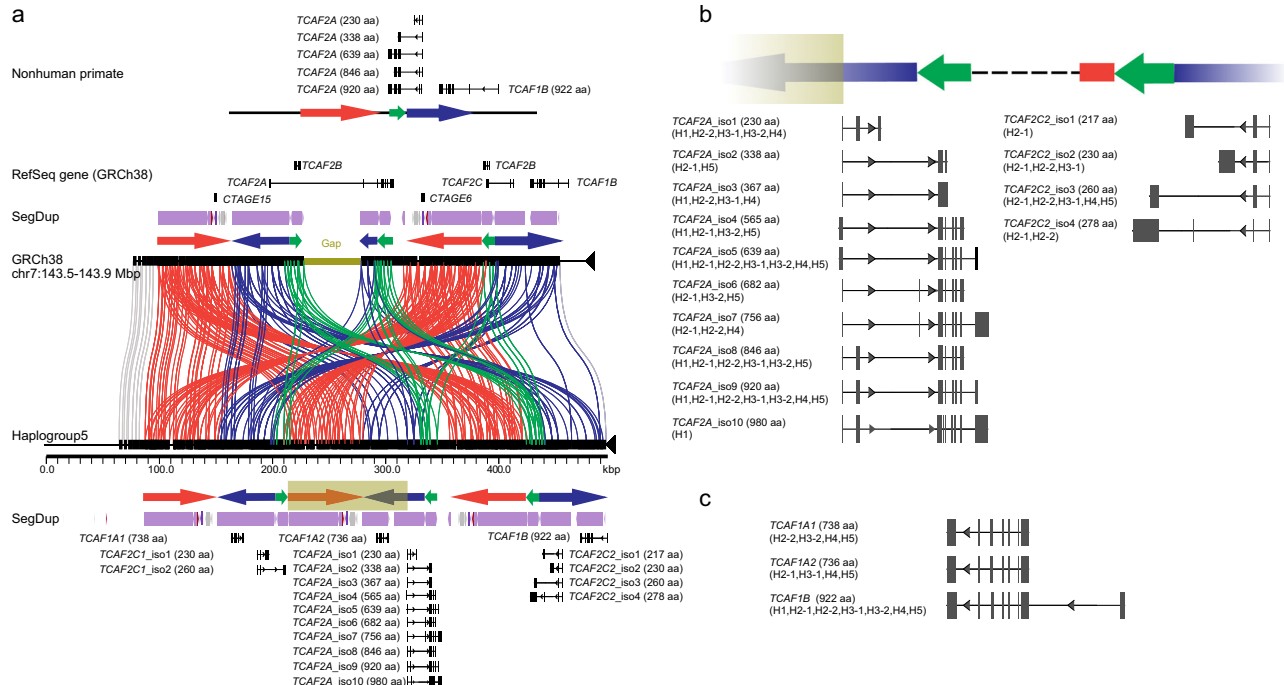

**Fig. 3 *TCAF* gene and isoform diversity among human haplotypes. a** Miropeats analysis reveals great consistency between Haplogroup 5 and the gapped 7q35 *TCAF* locus in the human reference genome (GRCh38). Colored arrows are annotated *TCAF* SDs and lines connecting the sequences show regions of homology. The dark green rectangle in the sequence-resolved, gap-free Haplogroup 5 contig (bottom) represents the region missing in the GRCh38 sequence (dark green bar, top). Additional annotations include a schematic of *TCAF* sequence structure and gene models/isoforms for nonhuman primate, RefSeq gene track (GRCh38), segmental duplication (SegDup) tracks, and predicted *TCAF* models and isoforms using full-length non-chimeric transcripts from six human tissues (Methods). Note that the *TCAF2A* gene model in GRCh38 is incorrect due to the presence of the gap in the middle of the *TCAF* SD region. For illustration, all predicted gene models and isoforms are aligned to Haplogroup 5 sequence for **b** *TCAF2A* and **c** *TCAF2C*, along with annotations for amino acid (aa) sequence lengths and haplogroups in which they are observed. For simplicity, we skip sequences between *TCAF2A* and *TCAF2C2* (dashed line). Detailed haplotype-specific and/or tissue-specific gene models and isoforms can be found in Supplementary Figs. 6–13 and Supplementary Data 5.

showed that this group is more closely related to Haplogroup 3 and likely a result of a deletion event over Haplogroup 3 (Supplementary Fig. 10 and Supplementary Data 3).

The breakpoints of these six large structural differences map within nearly identical (>99.5%) SDs, consistent with the action of non-allelic homologous recombination (NAHR) (Supplementary Data 3). By identifying the longest, high-identity sequences around the putative breakpoints (Supplementary Data 3), we show that four associate with DupA SDs, while two maps around DupC SDs. For example, the inferred inversion breakpoints of Haplogroup 4 are immediately flanking two inversely oriented, nearly identical sequences within DupC SDs (~39.6 kbp; sequence identity > 99.8%). In addition, our analysis of sequence composition reveals the presence of nearly identical LINE/SINE elements in sequences flanking the putative breakpoints (Supplementary Data 3). The inferred breakpoints for the deletion event between Haplogroups 3-1 and 4 overlap with two perfectly identical (100%) 2.5 kbp *CTAGE* (Cutaneous T Cell Lymphoma-Associated Antigen 1) family sequences (Supplementary Fig. 8 and Supplementary Data 3). Remarkably, most of the deletion events are similar in size ranging in length from 129.7 to 134.4 kbp. Thus, our data highlight a rapid evolutionary process that involves large-scale duplication, deletion, and inversion events via NAHR but where deletion events are constrained.

**Discovery of *TCAF* genes/isoforms**. The long-read BAC haplotypes reported here allow us to revisit *TCAF* annotation. Functional studies have shown that *TCAF1/TCAF2* are highly

expressed in the prostate, brain, esophagus, prostate, and skin tissues[12] (GTEx Portal, https://gtexportal.org). To this end, we targeted capture of FLNC *TCAF* transcripts and generated 480,700 FLNC transcripts from six human tissues, including dorsal root ganglion, esophagus, fibroblast, skin, fetal brain, and testis, as well as 50,885 FLNC transcripts from a chimpanzee lymphoblast cell line (Supplementary Data 4). *TCAF* transcript on-target rates range from the lowest in the testis sample (2.2%) to the highest for the chimpanzee lymphoblast sample (15.5%). To identify *TCAF* models, we aligned high-quality FLNC transcripts to the assembled haplogroup sequences and only retained transcripts that have complete open reading frames with >100 amino acids (aa) in length (Methods). Of note, we followed the previously described nomenclature[8] to delineate *TCAF* models and isoforms. Overall, our analysis indicates the expression of *TCAF* genes in a wide variety of tissues and reveals diverse *TCAF* paralogs as well as isoforms (Fig. 3 and Supplementary Figs. 5–13; Supplementary Data 5). In particular, we recovered only single isoforms for individual genes in the *TCAF1* subfamily. In contrast, the *TCAF2* subfamily shows much more diversity with 10, 2, and 4 isoforms being identified for *TCAF2A*, *TCAF2C1*, and *TCAF2C2*, respectively. With respect to the current annotation in the human reference genome (GRCh38), all haplotypes confirm that *TCAF2A* (or *TCAF2* in RefSeq Release 200) is incorrectly annotated due to the presence of a gap at this locus (Fig. 3). The gene models and isoforms reported here represent the most correct and comprehensive annotations for the *TCAF* families (Supplementary Data 5).

Compared to the chimpanzee, every additional copy of an SD cassette identified among human haplotypes is associated with an additional copy of *TCAF1A* and *TCAF2C* paralogs, although they are incomplete copies to *TCAF1B* and *TCAF2A*, respectively (Fig. 3 and Supplementary Figs. 4–12). Specifically, all *TCAF1A1/TCAF1A2* paralogs match to the last seven exons of *TCAF1B*, while the *TCAF2C1/TCAF2C2* paralogs consist of the first three exons of *TCAF2A* (Fig. 3). In addition, the predicted breakpoints between Haplogroup 2-2 and Haplogroup 5 coincide with the *TCAF1A1* and *TCAF1A2* paralogs. The sequence alignment analysis shows that the breakpoint at Haplogroup 2-2 spans from the third to the fifth introns, and thus, the *TCAF1A1* copy of Haplogroup 2-2 is a fusion version of the *TCAF1A1* and *TCAF1A2* copies of Haplogroup 5 (Supplementary Fig. 8). Because the amino acid sequences encoded from these three paralogs are 100% identical, this suggests the actual breakpoints are within one of the three introns. Finally, the 100 kbp inversion between Haplogroup 4 and Haplogroup 5 raises the question whether the large-scale structural variation event has any effect on *TCAF* coding sequences. Notably, the inferred inversion breakpoints map to the second introns of *TCAF2A* and *TCAF2C2*. There are only two differences in the *TCAF2A* exon alignment between Haplogroups 4 and 5: one is a synonymous change in the second exon, and the other causes a nonsynonymous change (R479Q) in the third exon, which is beyond the inferred breakpoints (Supplementary Fig. 13). Thus, our results predict that this inversion shuffles the last six exons of *TCAF2A*, which were ligated with the first two exons of different *TCAF2A/TCAF2C2* paralogs on the two haplotypes without disrupting or altering the coding sequence potential of *TCAF2A/TCAF2C2*.

It should be noted that although the structural changes largely increase and decrease the dosage of individual *TCAF* family members with relatively few predicted amino-acid differences, there is evidence that structural mutations and gene conversion events are likely affecting the regulatory landscape. We searched for candidate sites of interlocus gene conversion (IGC) among paralogs (Supplementary Fig. 14) and observed an IGC site that overlaps with a strong H3K27Ac signal (left white box, Fig. 1a) an epigenetic signature for enhancers. Using paralog-specific copy number genotypes (Methods), we found a reduction in copy number at this locus (i.e., the acceptor site; chr7:143,615,002–143,624,482, GRCh38; the left white box, Fig. 1a and Supplementary Fig. 15), which, interestingly, is significantly negatively correlated with an increase in copy number at its paralogous locus (the donor site; chr7:143,833,163–143,842,658; the right white box, Fig. 1a and Supplementary Fig. 15) (Pearson's correlation = $-0.3$, $p$ value = $5 \times 10^{-16}$) in multiple continental populations (Supplementary Fig. 16). This result is consistent with the hypothesis of an IGC event copying the donor sequence over the acceptor locus, resulting in reciprocal copy number changes at these two loci. Notably, we find that this paralog-specific copy number at the donor locus is negatively correlated ($R = -0.18$, $p$ value = $3.6 \times 10^{-7}$) with the (absolute) latitudinal location of human populations while the relationship for the acceptor site is positive ($R = 0.23$, $p$ value = $2.5 \times 10^{-11}$; Supplementary Fig. 17). While the observed correlations between latitudes of individual populations and paralog-specific copy numbers at the IGC sites may be the result of natural selection, such strong correlations are relatively common among similar loci across the genome ($p$ value = 0.22; 1,000 random SD loci of genomic shuffling) (Supplementary Fig. 18), suggesting demographic history among populations also contributes to this observation. However, only seven of the 1,000 sampled SD loci have correlation coefficients >0.23 and remain significant after Bonferroni's correction (Supplementary Fig. 19). To further

investigate if population structure in the data confounds with the observed correlation, we conservatively fit a regression model between the paralog-specific *TCAF* copy number and the latitudinal location of human populations and include the first 10 principal components as covariates, which collectively account for 90.8% of the variance (Methods). Consistent with the random sampling analysis, the correlation $p$ value becomes insignificant ($p$ value = 0.14; Supplementary Fig. 20), indicating that the observed correlation is likely due to the evolutionary relationship among populations.

**Evolution of *TCAF* segmental duplications.** To reconstruct the evolutionary origin of the *TCAF* haplotypes in modern humans, we leveraged these high-quality human and nonhuman primate haplotypes for phylogenetic reconstruction (Table 1 and Supplementary Data 6). The mean sequence identities for DupA, DupB, and DupC SDs are high among human haplogroups (99.80% [s.d.: 0.04], 99.71% [s.d.: 0.10], and 99.85% [s.d.: 0.05], respectively) (Supplementary Fig. 21), while the sequence divergence between human and nonhuman primate *TCAF* haplotypes are compatible to published whole-genome estimates (Supplementary Fig. 21)[26,27]. Together, these findings are consistent with a recent *TCAF* SD expansion or IGC. We considered each of the duplicated segments independently guarding against confounders such as IGC and NAHR events. Because several SDs represent hybrids of two SD paralogs due to NAHR fusion events (Fig. 2), for example, we restricted our analysis to the larger segments of hybrid SDs. To minimize the effects of IGC (Supplementary Fig. 14)[8], we applied GENECONV (v1.81a) to exclude sites corresponding to 80.5%, 61.7%, and 70.6% of DupA, DupB, and DupC sequences, respectively (Supplementary Data 7). Phylogenetic reconstruction and dating estimates on this filtered set showed that all human copies form a single clade with a 100% posterior support and coalesce at 1.06 (95% C.I.:0.87–1.26), 1.72 (95% C.I.: 1.26–2.21), and 1.47 (95% C.I.: 0.74–2.54) million years ago (Mya), for DupA, DupB, and DupC SDs, respectively (Fig. 4a and Supplementary Figs. 22–24). While a few differences in topology were noted among the inferred phylogenies (Supplementary Figs. 22–24), these estimates, with the exception of DupB, overlapped coalescent estimates obtained from a 12.3 kbp single-copy unique region (Fig. 1a, the middle green segment at the bottom), which suggests that the two supergroups: H5 (Haplogroup 5, Haplogroup 2-1, and Haplogroup 2-2) and H4 (Haplogroup 4, Haplogroup 3-1, Haplogroup 3-2, and Haplogroup 1) diverged from each other ~0.74 Mya (kya; 95% C.I.: 0.46–1.03) (Supplementary Fig. 25a). Importantly, these findings suggest that the duplications began to occur before humans and archaic hominins diverged.

To further refine our inferences on the evolution of *TCAF*, we realigned genome sequence data from four archaic hominins along with human and chimpanzee samples to a custom, gapless human reference chromosome 7 using GRCh38 and sequence-resolved BAC haplotypes to generate a joint single-nucleotide variant (SNV) call set (Methods). Due to the limitations of short-read data, we restricted this analysis to three unique regions (52.3 kbp in total) flanking and internal to the *TCAF* locus (Fig. 1). We identified 1,275 SNVs (QV > 20) and used these to computationally phase and to construct haplotypes from 802 samples (Supplementary Figs. 2 and 24). Using these archaic hominin haplotypes with the sequences of the seven BAC haplogroups, we estimate the time to the most recent common ancestor (TMRCA) of all hominins being 0.77 Mya (95% C.I.:0.55–1.03 Mya), which is consistent with that reported above. We should note that these time estimates are subject to uncertainties, such as mutation rate, and therefore caution is

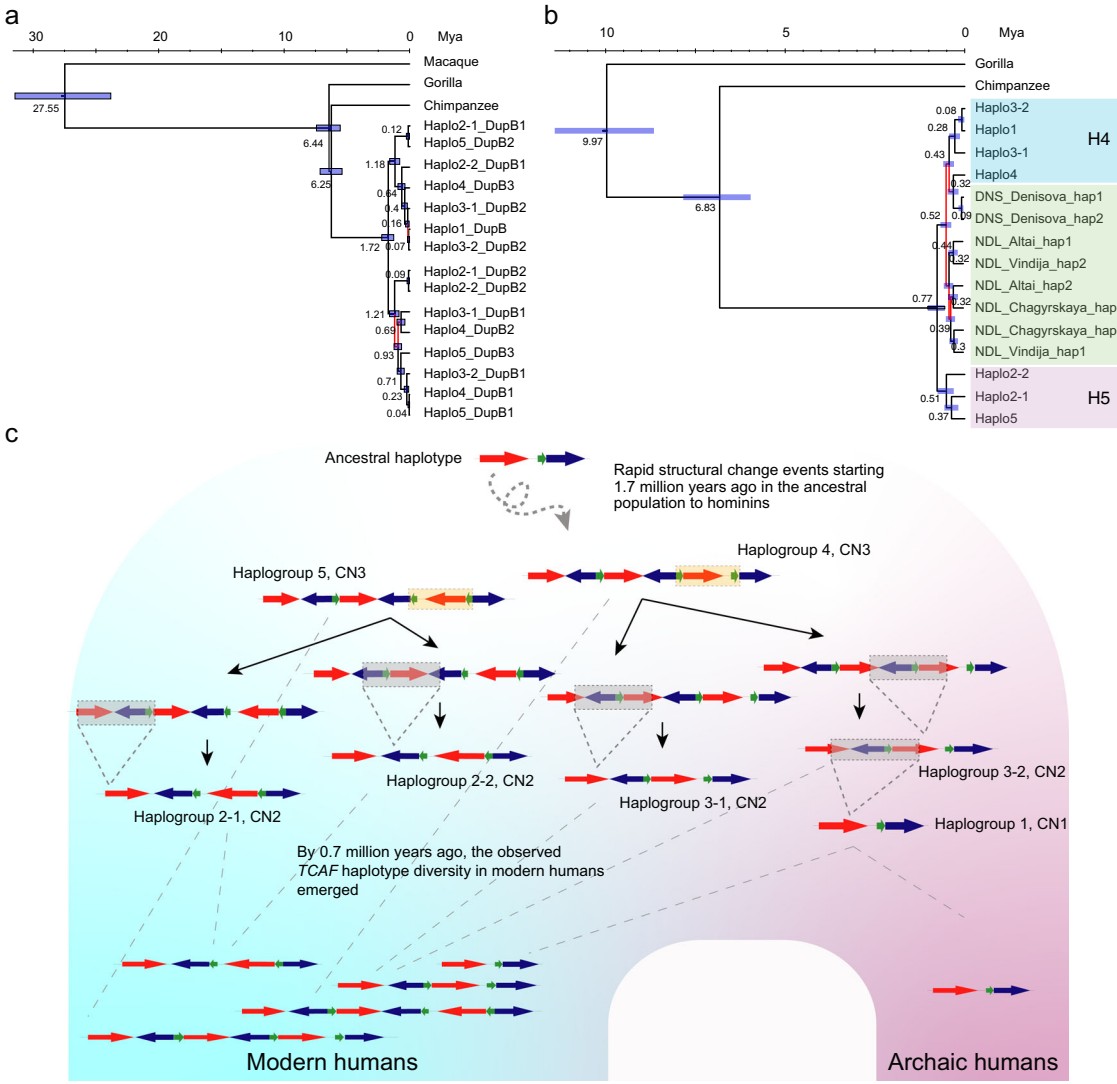

**Fig. 4 Evolutionary reconstruction of *TCAF* structural diversity. a** Phylogeny of the haplogroups was inferred using *TCAF* DupB sequences and BEAST (v.2.6.2) with five independent runs of 10 million iterations of Markov Chain Monte Carlo (Methods). Numbers and horizontal bars at internal nodes indicate point estimates and 95% highest-posterior density intervals for the divergences (in million years ago, Mya), respectively. Branches with posterior probabilities <90% are colored in red. See Supplementary Figs. 22–24 for results of other SD sequences. **b** Inferred phylogeny of the modern human haplogroups and archaic hominin haplotypes using the 12 kbp unique sequences embedded within *TCAF* SDs (Fig. 1). Haplotypes of archaic samples were generated using high-confident single-nucleotide variants (SNVs) called within the unique diploid region. Phylogenetic inference was performed similarly as described above. **c** Schematic model for the evolution of *TCAF* haplotypes in humans based on phylogenetic inferences (Fig. 4a, b and Supplementary Figs. 22–24). Colored arrows are *TCAF* SDs; orange and gray areas indicate relative inversion and deletion events between haplogroups, respectively. Short dashed lines indicate putative breakpoints of structural changes between haplogroups, while the long-dashed lines illustrate lineage sorting.

warranted when interpreting these dates (Methods). Nevertheless, multiple phylogenetic methods (Fig. 4b and Supplementary Fig. 25) show that all archaic haplotypes and the supergroup H4 are more closely related than either is to the supergroup H5, and the clade of archaic and supergroup H4 haplotypes coalesces ~0.53 Mya (Fig. 4b). Thus, our results show that as early as 1.72 Mya the *TCAF* locus (DupB SD) began to duplicate and that by 0.74 Mya most of the *TCAF* haplotype diversity that we observe in modern-day humans had already emerged. Archaic hominins, in contrast, show a reduction in genetic diversity without the associated copy number duplication diversity found in modern humans (Fig. 4c).

**Hominin diversity and natural selection of *TCAF* haplotypes.** We used the combined haplotypes constructed from the 53.2 kbp of unique regions to further explore genetic diversity among

archaic and present-day hominins. A haplotype-based principal component analysis (PCA; Methods) reveals distinct clusters (Supplementary Figs. 27–28) corresponding in part to the two human haplogroups H4 and H5 identified from the BAC sequence analysis (Supplementary Fig. 28). Using a supervised clustering method based on haplotype PCA and machine-learning algorithm t-SNE[28] for dimensionality reduction and visualization (Methods), we show that contemporary human haplotypes can be classified into 12 different clusters (Fig. 5a and Supplementary Fig. 29) with an estimated haplotype heterozygosity of 0.885 (Methods), which is much higher than the reported genome-wide estimated mean of 0.534[29]. Phylogenetic reconstruction using 10 haplotypes drawn at random from each cluster reproducibly shows a maximum likelihood tree closely resembling this cluster-based relationship (Fig. 5b). Considering the two flanking and unique regions independently also confirms

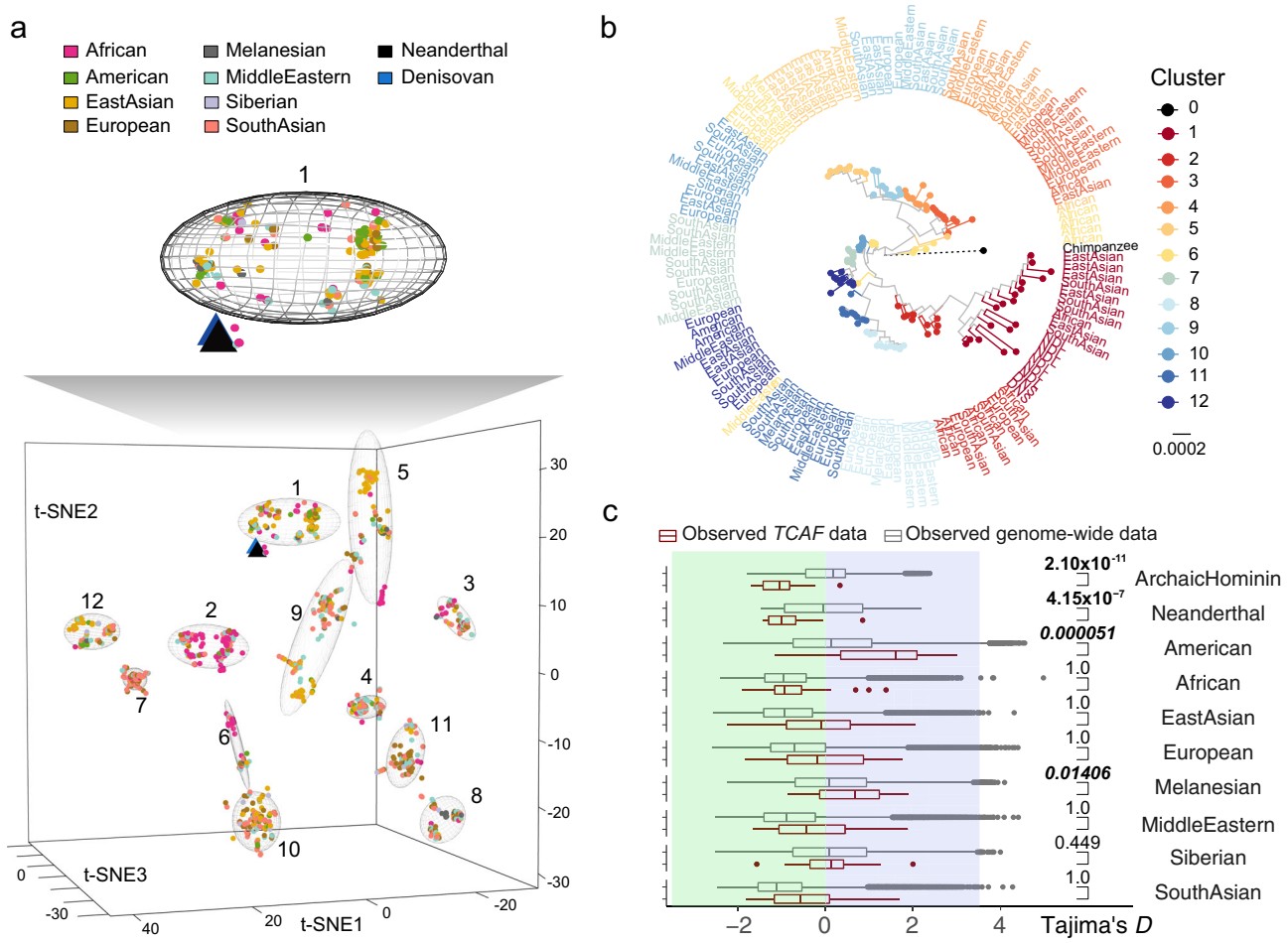

**Fig. 5 Archaic hominin versus human haplotype diversity.** Haplotypes were inferred using 1,275 SNVs in the three unique diploid sequences around the *TCAF* SD region (Fig. 1). **a** Haplotype-based principal component analysis (PCA) was performed, followed by haplotype clustering and cluster visualization using t-SNE—a dimension-reduction technique (Methods). On the t-SNE plot, each dot/triangle is a haplotype and colored according to population/species origin. Neanderthal and Denisovan haplotypes are indicated by the black and blue triangles, respectively. Numbers and ellipses in the 3D t-SNE plots indicate individual clusters (Supplementary Figs. 27–32). The zoom-in above the 3D t-SNE shows that all archaic haplotypes are in close proximity to each other and associate with cluster 1. **b** The maximum likelihood phylogeny was constructed using 10 randomly selected haplotypes from the 12 inferred clusters, in addition to eight archaic and one chimpanzee haplotypes. Note that the branch length of chimpanzee (dashed line) is truncated by 90% of its actual length for the purpose of illustration. **c** Comparisons of the Tajima's $D$ distributions between the *TCAF* locus and the entire genome in human and archaic populations. Tajima's $D$ statistics are computed for individual archaic and modern groups across the entire genome (gray) versus *TCAF* (red) based on 2 kbp windows. For each boxplot, the lower and upper hinges correspond to the first and third quartiles, respectively, in addition to the median value. The upper and lower whiskers show 1.5 * IQR from the upper and lower hinges, respectively, where IQR is the inter-quartile range defined as the distance between the first and third quartiles. $P$ values (one-tailed) are empirically determined by testing if the observed values at *TCAF* (29 windows) are significantly more negative (green) than those from the genome (1,224,349 windows), except for the American and Melanesian groups, where their $p$ values indicate if the observed $D$ values at *TCAF* are significantly more positive (italic) than the those from the genome. Bold fonts indicate Bonferroni's $p$ values (one-tailed) < 0.05.

a phylogeny of 12 distinct clusters with different BAC haplogroup representing different clusters (Supplementary Figs. 29–30). Among humans, no single population is restricted to a specific cluster, although some clusters appear enriched. For example, clusters 2 and 7 largely consist of African (60.3%) and South Asian (50.6%) haplotypes, respectively (Supplementary Fig. 31). Assuming an equal split of the overall diploid copy number estimates between the two haplotypes in an individual, we found significant differences in *TCAF* copy number among haplotype clusters (Bonferroni corrected $p$ values < 0.05; Mann–Whitney $U$ tests) (Supplementary Fig. 32).

In sharp contrast to the extensive haplotype diversity in modern human samples, the eight archaic hominin haplotypes are virtually identical belonging to a single homogenous group (Fig. 5, Supplementary Figs. 27–32). Under a panmictic model

and assuming an ancestral hominin origin of ~0.74 Mya for *TCAF* haplotypes, it is highly unlikely that all eight archaic haplotypes would form a single cluster given modern-day human diversity ($p$ value = $2.1 \times 10^{-7}$, $100 \times 10^6$ permutations). To formally test whether natural selection plays a role in these unusual patterns in both modern and archaic humans, we performed a selection test using the Tajima's $D$ statistic and coalescent simulations under plausible demographic models to assess significance (Methods). In several human groups, we noticed that the Tajima's $D$ values at *TCAF* significantly deviate from those across the whole genome (Fig. 5c). Specifically, the Tajima's $D$ values of the archaic hominins are significantly negative compared to the rest of the genome (median Tajima's $D < -0.99$; Bonferroni's $p$ values $\leq 4.15 \times 10^{-7}$), regardless of the presence of the Denisovan individual in the test. In contrast,

while the *TCAF* Tajima's $D$ distributions among most modern human groups are compatible with those of the genome, there are two notable exceptions: Native American (median Tajima's $D = 1.62$; Bonferroni's $p = 5.1 \times 10^{-5}$) and Melanesian (median Tajima's $D = 0.69$, Bonferroni's $p = 0.01406$) samples (Fig. 5c).

To account for biases in the Tajima's $D$ values due to demographic effects, we simulated data using multiple population-specific models published recently (Supplementary Data 9; Methods) and computed Tajima's $D$ values across 2,000 bp windows. Overall, the simulated Tajima's $D$ distributions qualitatively recapitulate the observed data; however, in all cases, we note differences in individual populations (Supplementary Fig. 33). These differences are likely due, in part, to the presence of multiple geographically closed populations in each supergroup, and, also in part, due to different population samples underlying the demographic models used in our study. Consistent with the results of the comparison between the *TCAF* locus and the whole genome, in both archaic hominin groups, we observe significantly negative Tajima's $D$ values from the data compared with those from the simulations (Bonferroni's $p$ values $\leq 6.62 \times 10^{-6}$; Supplementary Fig. 34). Within modern humans, no populations show significantly negative Tajima's $D$ values compared with simulated data though we do note that the Native American samples show significantly more positive Tajima's $D$ values than those based on simulations (Bonferroni's $p$ value $= 3.32 \times 10^{-11}$; Supplementary Fig. 34), possibly as a result of balancing selection at this locus in this population.

To further investigate if the observed negative Tajima's $D$ values in the archaic hominins are more likely to be driven by positive selection, rather than purifying selection resulting in a reduction of genetic diversity, we computed the Fay and Wu's $H$ statistic[30]. Briefly, a negative $H$ value indicates an excess of high-frequency derived alleles at the locus of interest—a signature of positive selection but not purifying selection (Methods). Consistent with the Tajima's $D$ results, we observe significantly more negative Fay and Wu's $H$ values from the *TCAF* locus than those from the genome-wide data in both the archaic hominin and Neanderthal groups (Supplementary Fig. 35). We also examined and found that the simulated Fay and Wu's $H$ distributions are consistent with the observed data across all hominin groups, except for a bias in the American (Supplementary Fig. 36). While both the Neanderthal and archaic hominin groups are qualitatively negative ($H \leq -0.86$), only the archaic hominin group is significant compared to the simulated distribution (Bonferroni's $p$ value $= 0.00917$; Supplementary Fig. 37). The smaller sample size of the Neanderthal group ($n = 3$) is likely limiting the power of the $H$ statistic. In addition, we note that while the simulated $H$ distribution under neutrality is centered around zero as expected, its distribution is heavily left-skewed (Supplementary Fig. 37), suggesting that the critical $p$ values computed in our tests are relatively conservative. Of note, for both the Neanderthal and archaic hominin groups, the joint density of these two statistics from the observed *TCAF* data largely falls within the negative range, which is distinctively different from that of simulated data (Supplementary Fig. 38) and consistent with a signature of positive selection and unlikely explained by the demographic history of these hominins.

Not only are the distributions of the two statistics for the *TCAF* locus significantly different from the neutral expectations for archaic hominins, we also find that multiple *TCAF* windows fall in the extreme (left/negative) tails of both the Tajima's $D$ and Fay and Wu's $H$ genome-wide distributions for both observed and simulated data (Supplementary Data 10), providing evidence for positive selection. In contrast, the Native American results map once again to the (right/positive) tails of the observed and simulated distributions but, instead, show significant positive

Tajima's $D$ values consistent with the signature for balancing selection (Supplementary Data 10). Because these windows having values that are unexpected under neutrality, they provide further evidence for selection. Taken together, our results suggest distinct signals of selection in both humans and archaic hominins.

These dramatic differences in diversity and potentially distinct selection forces pose the question of whether there are any functional differences among the different *TCAF* haplogroups. Because our structural analyses suggested dosage and regulatory differences among common haplogroups (Supplementary Figs. 31–32), we investigated *TCAF2* expression using the GTEx multi-tissue data (release v8) and Human Protein Atlas (HPA; https://www.proteinatlas.org). Based on the BAC references, we identified four tagging SNVs in the unique diploid region embedded among the *TCAF* SDs that could distinguish super haplogroups H4 and H5 (Fig. 6). Because the two super haplogroups structurally differ by a 100 kbp inversion, we expect a reduction of homologous recombination events between the two groups and therefore an increase in linkage disequilibrium (LD) in hominins. As expected, we observe substantial LD across all three unique diploid regions in all archaic and modern samples and confirm nearly complete LD among the four tagging SNVs ($D' > 0.98$, $R^2 > 0.89$) (Supplementary Fig. 39). Using these data, we again find that all archaic samples are fixed for the H4 supergroup and infer the frequencies of H4, H5, and other forms in modern humans being 57.1%, 40.3%, and 2.6%, respectively. While the H4 haplogroup is found at high frequency among non-Africans, especially in Melanesians (72%), some Native Americans (e.g., the Karitiana, >72%), and Northeast Asians (e.g., the Mongolians, >68%), the H5 group segregates in higher frequency among Europeans (e.g., the Basques, >81%), Middle Easterners (e.g., the Druze, >61%), and some Native Americans (e.g., the Surui, >83%) (Fig. 6a). Consistent with our haplotype-based PCA clustering results, we observed high frequencies of the H4/H5 heterozygous form (orange in Fig. 6b) across populations, with the highest found in Southeast Asians and Native Americans (>60%) and the lowest in some Melanesian and African populations (<20%).

Given the common presence of these two super haplogroups in humans (Fig. 6a, b), we collected multi-tissue eQTL data from the GTEx Project (release v8) and found that all four tagging SNVs are *TCAF2* eQTLs (Fig. 6c). Among tissues, the H5 haplogroup (alternative) alleles exclusively increase the *TCAF2* expression in liver and prostate tissues ($p$ values $< 1 \times 10^{-5}$), while the H4 haplogroup (reference) alleles are significantly associated with increasing *TCAF2* ($p$ values $< 1 \times 10^{-10}$) expression in tissues, such as thyroid and tibial nerve tissues, in addition to others (Fig. 6c). In addition, using the GTEx data and the HPA database, we find that both *TCAF2* and *TRPM8* are highly expressed in the liver and prostate (Supplementary Fig. 40). To further examine the expression of *TCAF2* and *TRPM8* across different cell types in liver and prostate tissues, we used the single-cell RNA-seq data from the HPA (Supplementary Figs. 41–42). Both *TCAF2* and *TRPM8* are almost exclusively expressed in clusters (clusters c-0, 2, 5, 7, and 11) corresponding to hepatocytes of the liver and in the prostate glandular (clusters c-6 and 13) and endothelial cells (cluster c-7). These findings together imply that the difference in *TCAF2* expression between the H4 and H5 haplogroups may play a role in differentially regulating *TRPM8* in the liver and prostate tissues among human groups.

## Discussion

Genetic changes in SDs are now well recognized as an important source of functional innovation in human evolution due, in part,

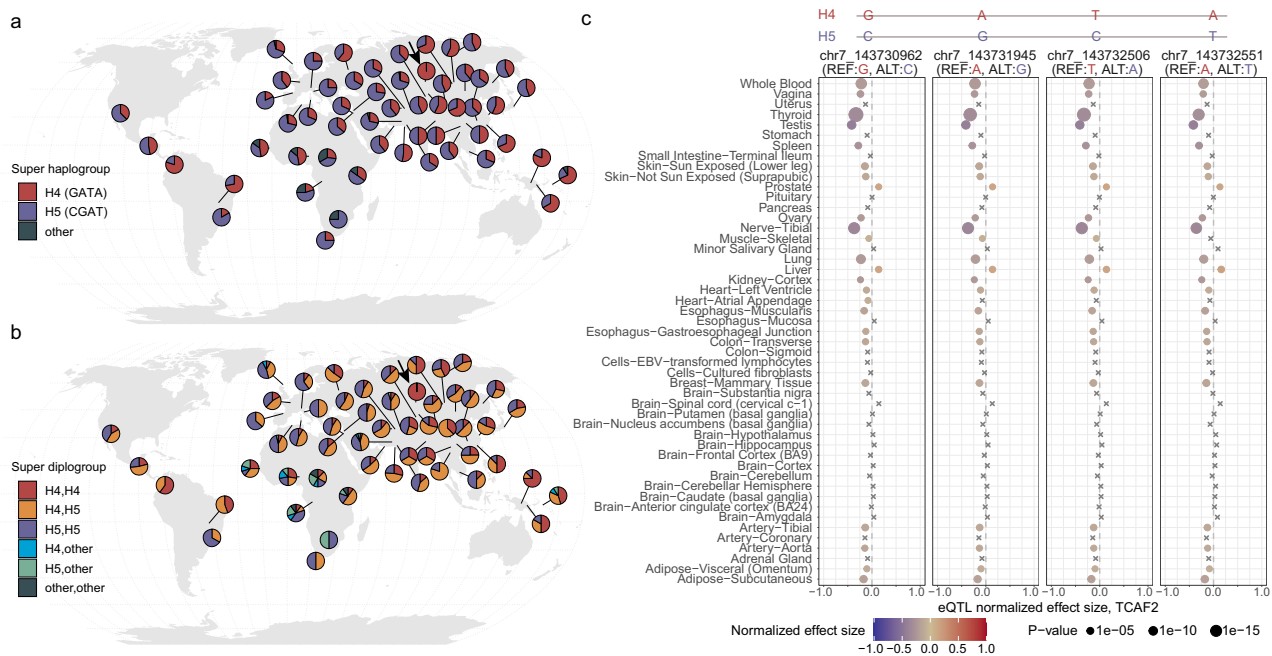

**Fig. 6 H4 and H5 super haplogroup distribution and eQTL analysis.** Four tagging SNVs were first identified by perfectly separating super haplogroups H4 and H5 among the seven BAC haplogroups and confirmed based on patterns of linkage disequilibrium (LD) in samples from the HGDP panel. **a**, **b** Distributions of super haplogroups H4 (red, 57.1%) and H5 (blue, 40.3%) and their diploid type in the HGDP populations across the world. Other haplotypes were found in ~2.6% of the samples. Note that archaic hominin haplotypes from the Neanderthal ($n = 3$) and Denisovan ($n = 1$) samples all carry H4 haplotypes, and as a representation the geographic location of these samples was placed in the Altai Mountains in Siberia (arrows). These maps were generated using the R packages rgdal (v1.5), scatterpie (v0.1.6), and Natural Earth. **c** Multi-tissue eQTL plots show consistent patterns of associations between the four tagging SNVs and expression levels of *TCAF2* across 50 tissues. Effect sizes were calculated as the effect of the alternative allele (blue) relative to the reference allele (read) as defined in GTEx (release v8) and are scaled using color. The (unadjusted) *p* values of eQTL association are computed by testing the alternative hypothesis that the slope of a linear regression model between genotype and expression deviates from 0 and represented by dot sizes. Note that crosses (×) indicate insignificant associations.

to subsequent rounds of duplication, deletion, or inversion[13,14]. Such regions, however, are often frequently overlooked as part of comparative studies, association studies, or scans of selection because of their structural complexity and high degree of homology. In this study, we first delineate the complex structural diversity of the *TCAF* locus in humans and compare it to non-human primates at the sequence level in order to reconstruct its evolutionary history[12]. Of note, the complete assembly of 15 haplotypes by tedious BAC-based sequencing was sufficient to capture more than half of the genetic diversity of *TCAF* SDs in humans. Based on these data, we develop a model of rapid diversification with respect to both haplotype and *TCAF* SD copy number diversity (Figs. 1–3). It is worth noting that despite the recent advances in long-read sequence assembly, the most complex haplotypes still remain collapsed and unassembled in some of the recently released human genomes[31–33] (Supplementary Fig. 4 and Supplementary Data 2). We believe that this is due partly to the read-length limitation of the data as well as the near-perfect sequence identity among *TCAF* SDs that lead to ambiguous read partitions during haplotype assembly. This highlights the importance of alternate approaches, such as large-insert BAC clones, to construct highly accurate haplotype-resolved assemblies. More sophisticated assembly approaches and long-read technology for generating longer and even more accurate sequence reads will likely make such investigations much more routine in the future[34].

The structural complexity and copy number variation of this region underscore its mutational dynamics. The high degree of sequence identity among *TCAF* paralogs (>99.7%), their clustered and tandem orientation, and their large size (10–60 kbp) promote recurrent structural rearrangements making the entire locus genetically unstable as a result of NAHR[13]. Human Haplogroups 4 and 5, in particular, carry more *TCAF* SD paralogs than others, and, thus, both are predicted to be subject to a higher likelihood of structural mutation. Our analysis predicts that H4 would be more predisposed to structural rearrangements than H5 due to a greater number of directly orientated duplications (DupA and DupB) (Fig. 2). This prediction is consistent with the observation of more subgroups within supergroup H4 in the haplotype-based PCA (Supplementary Figs. 27–30). Surprisingly, this predisposition to recurrent structural diversity is specific to modern-day humans as our sampling of eight archaic hominin haplotypes suggests a static H4 haplogroup without a single duplication event. Given our estimates of the origin of the duplications (~1.7 Mya) and haplogroup coalescence (~0.77 Mya), the lack of diversity in the Neanderthal and Denisovan lineages is highly unlikely.

One of the challenges studying this region is the wide-spread IGC we identified among the *TCAF* SDs. IGC essentially erases signatures left by previous mutation events in the acceptor site, thus biasing towards younger TMRCA estimates[35,36]. While we mitigated such a bias by focusing our analysis on non-IGC sequences, this reduced the available dataset limiting our power to infer its evolution at a finer level of resolution. Interestingly, the analysis uncovered a significant correlation between paralog-specific copy number and the latitudinal location of human populations precisely over a potential enhancer region, suggesting that IGC may have contributed to regulatory differences of *TCAF* genes among human populations. Such co-option of IGC has been noted before among duplicate genes. For example, a human-

specific, pseudogene-mediated IGC of *SIGLEC11* replaced the 5′ upstream region and coding exons of *SIGLEC11* by its adjacent pseudogene *SIGLEC16P* in the human lineage. This event has been hypothesized as adaptive during human brain development[37]. While it is likely that the observed correlation is confounded by the effects of demography, given the importance of *TCAF* in thermal regulation and sensing, it remains tempting to hypothesize that IGC contributes to local adaptation by changing dosages through rapid replacement of regulatory sequences in addition to SD.

Our more complete reference genomes coupled with targeted cDNA sequencing also provide a comprehensive *TCAF* annotation for humans and nonhuman primates (Fig. 3 and Supplementary Figs. 4–12). We show that each additional *TCAF* SD cassette (DupA, DupB, and DupC) generates additional copies of *TCAF1A* and *TCAF2C* through incomplete gene duplication. Gene duplication has been known to be an important source of genetic novelty and adaptive evolution[1,38], especially for the great ape lineages given the surge of duplications predicted to have occurred in the hominid ancestor[39]. Throughout the evolution of mammals, *TCAF* genes have been highly conserved, and human is the only lineage that is known to carry additional copies. The coding sequences among the individual paralogs are highly conserved as a result of their recent origin arguing for the relative importance of *TCAF* dosage differences in regulating the TRPM8 ion channel as suggested by a previous study[12]. We also observe, however, considerable isoform diversity as a result of the incomplete nature of the SD particularly among *TCAF2* paralogs suggesting a potentially more complex relationship with the ancestral genes. This pattern of incomplete duplication resembles the evolution of the neural human-specific duplication *SRGAP2* genes[5], where the younger, incomplete duplicated paralog *SRGAP2C* inhibits the function of the ancestral copy *SRGAP2A*, and in mice experiments this in turn prolongs the maturation of synapses and subsequently increases the density and the length of synapses[4]. Taken together, our results bolster the hypothesis of recent incomplete gene duplications and adaptive human evolution.

Remarkably, our data support a model of two distinct forces of natural selection possibly operating on the same locus over the last half million years of hominin evolution. We propose that diversifying or balancing selection is likely acting in at least some human populations, particularly out-of-African populations such as Native Americans, to maintain and expand haplotype and structural diversity. While the exact mechanism and phenotype that balancing selection are acting upon are not clear, the differential eQTL signal related to *TCAF2* expression between supergroup H4 and H5 alleles in multiple tissues would be consistent with antagonistic pleiotropy, where two haplotypes have opposing effects on two different traits depending on either environmental or life-history conditions[40,41]. In contrast, Neanderthal and Denisovan show a paucity of genetic variation, and while the sample size is still limited, this observation is unlikely to change with the sequencing of additional archaic genomes. The negative Tajima's *D* and Fay and Wu's *H* values at this locus compared with those of the genome-wide and simulated values, which accounts for differences explained by demographic history, suggest that the observation of the single-haplotype without duplication in the archaic samples is unlikely under neutrality. Using coalescent simulations to assess the significance of a population genetics test is known to be more reliable than choosing outliers as candidates because these simulations explicitly account for effects of population demographic history and genomic variation in mutation and recombination rates[42,43]. However, we also note that our selection inference is limited by our understanding regarding the demography of these archaic

hominins and the underlying models and should be further tested in the future by incorporating new archaic samples and demographic models. Nevertheless, given our inference results, we hypothesize that positive selection has reduced genetic diversity at the *TCAF* locus in these archaic hominin lineages.

The opposing effects on the *TCAF2* expression found between H4 and H5 haplogroups and the observations of high expression of both *TCAF2* and *TRPM8* in liver and prostate tissues are particularly intriguing. While the classical function of *TRPM8* is as cold and menthol receptors for sensory neurons, it was first identified as an overexpressed gene in prostate cancer[44,45] and is known for its role in controlling calcium flux and homeostasis—an important physiological process in cold signaling[46]. Moreover, recent studies demonstrated the role of *TRPM8* in energy homeostasis; specifically, it is required for precise thermoregulation in response to cold in the fed and fasting states in mice[45]. Not only is *TRPM8* required for cold detection in mice[47], but *TRPM8*-knockout mice raised at mild cold temperatures develop late-onset obesity and display a metabolic dysfunction with a higher content of lipid droplets in liver[48]. This implies that the TRPM8 ion channel, in addition to its role in detecting environmental cold, is a known regulator of insulin homeostasis through neuronal control of liver insulin clearance and, thus, required for precise thermoregulatory responses to cold and fasting[48,49]. Interestingly, a rare variant within the *TCAF* locus (chr7:143735931) was recently reported to be associated with type 2 diabetes in sub-Saharan African populations[50]. Given this variant's restricted geographic distribution and low frequency in the GTEx data (allele frequency: ~0.4%; GTEx portal as of 2020.07.21), it explains neither the observed patterns of diversity nor the selection and the eQTL signals observed here. Therefore, if the *TCAF* haplotype and structural diversity affect an individual's ability to regulate TRPM8 ion-channel gating and, thus, body temperature, we hypothesize that the antagonistic forces of selection promote adaptations to a cold environment or an enhanced ability to dynamically adapt to changing conditions, such as diet. Further experimental investigations on the adaptive significance and role of the different *TCAF* copies in regulating *TRPM8* in humans are required. These are now possible given that the structure and diversity of this dynamic locus has been more fully resolved.

## Methods

**Copy number genotyping in high-coverage Illumina short-read data.** To explore the temporal and spatial copy number variation of *TCAF* SDs, we leveraged a large collection of publicly available high-coverage Illumina genomes from 1,102 modern human samples, four archaic hominins, and 71 nonhuman great apes[17–23]. In short, sequencing reads were divided into multiples of 36-mer and then mapped to a repeat-masked human reference genome (GRCh38) using mrsFAST (v3.4.1)[51]. We allowed up to two mismatches per alignment to increase our mapping sensitivity and corrected for possible biases in read depth due to different levels of GC content in sequencing reads. Finally, the copy number estimate was computed by summarizing over all mappable bases at the locus of interest for each sample. Copy number distribution pie charts were generated using free maps downloaded from the Nature Earth (http://www.naturalearthdata.com/about/terms-of-use/), R package rgdal (v1.5), and scatterpie (v0.1.6).

**Library processing and assembly for *TCAF* BAC haplotypes.** We obtained the human BAC libraries used in this study from the Virginia Mason Research Center and the nonhuman primate ones from the Children's Hospital Oakland Research Institute. Probes for regions of interest were designed, radioactively labeled, and hybridized to the Performa filters, washed, exposed to Phosphor screens, and scanned on a Typhoon scanner. Positives are called and corresponding clones selected from the BAC library. We prepared barcoded libraries from clone DNA using Illumina-compatible Nextera DNA sample prep kits (Epicentre, catalog number GA09115) as described previously[52] and carried out paired-end sequencing (125 bp reads) on an Illumina HiSeq 2500. Reads were then mapped to the reference genome, GRCh38, to identify singly unique nucleotide k-mers (SUNKs)[16]. Non-overlapping BACs were pooled and sheared as described previously[53]. Libraries were processed using the PacBio SMRTbell Template Prep

kit following the protocol "Procedure and Checklist—20 kb Template Preparation Using BluePippin Size-Selection System" with the addition of barcoded adaptors during ligation. Up to ten barcoded libraries were then pooled at equimolar amounts and size-selected as a pool on the Sage PippinHT with a start value of 10,000–12,000 and an end value of 50,000. The resulting library was then sequenced on one Sequel SMRT cell 1 M by diffusion using Sequel v3.0 chemistry. We performed de novo assembly of pooled BAC inserts using Canu (v1.5). Reads were masked for vector sequence (pCC1BAC) and assembled with Canu, then subjected to consensus sequence calling with Arrow (v2.3.3). We reviewed PacBio assemblies for misassembly by visualizing the read depth of PacBio reads in Parasight (v7.6), using coverage summaries generated during the resequencing protocol. We performed BLAST (v2.10.1) pairwise alignment between the BAC sequences for the construction of haplotypes. Complete contigs for 15 *TCAF* haplotypes must satisfy the following two conditions: (1) overlapped DNA segment between BAC sequences must be at least 99.99% identical with no tolerated mismatch and (2) contig must anchor to unique location on both ends when mapping back to the human genome reference GRCh38.

**Identification of haplogroups and IGC tracts**. All sequence alignments in this study were performed using MAFFT (v7.453). To determine the structural differences among the haplogroups, we built single-base level sequence identity profiles using pairwise sequence alignment, followed by smoothing over windows of 500 bp with a step size of 100 bp. Because we were interested in recent, large IGC events, empirical IGC tracts are determined as more than two consecutive windows that have 100% sequence identity. To find additional support, we use a model-based program, GENECONV (v1.81a), to identify pairs of uninterrupted sequences with 100% sequence identity that are longer than expected given the overall pattern of variable sites in an alignment. To be conservative, we considered IGC tracts that have both simulation- and Karlin-Altschul- $p$ values < 0.05 reported by GENECONV.

To determine if the correlation between the latitudes of the populations and the paralog-specific *TCAF* copy numbers at the predicted IGC site, we performed a multivariate regression analysis that uses principal components (PCs) to account for the underlying population structure of the data (Supplementary Fig. 20). We applied the PCA using 715,881 genome-wide SNVs from the HGDP samples after linkage-disequilibrium pruning (removing variants with $R^2 > 0.4$ using the PLINK [v1.09][54]). We then conservatively fit a regression line that included the first 10 PCs as covariates, which collectively explains 90.8% of the variance, as follows:

$$\text{Paralog} - \text{specific copy numbers} \sim \text{latitude} + \sum_{i=1}^{10} PC_i$$

**FLNC transcripts for *TCAF* genes using long-read cDNA sequencing**. Total RNA was harvested from six human tissues (dorsal root ganglion: Clontech Laboratories, Inc., Takara [catalog #636150]; esophagus: Clontech Laboratories, Inc., Takara [catalog #636178]; fibroblast (Coriell); skin: Biochain Institute Inc. [catalog #R1234218-P], fetal brain: NCBI BioSample: SAMN09459150; testis: Clontech Laboratories, Inc., Takara [catalog #636533]) and one chimpanzee lymphoblast cell line. We purified polyA RNA using oligo-dT magnetic beads (Dynal: Thermo Fisher Scientific Inc.). Double-stranded cDNA with 96 random barcodes was prepared, amplified, and subjected to hybridization capture followed the protocols detailed in[55]. Hybridization probes were designed to target the duplicated exons of *TCAF2*, and sequences are given in Supplementary Data 11.

Following post-capture PCR, the amplified dsDNA was purified on magnetic beads (AMPure PB; PacBio) and then subjected to library preparation for long-read sequencing (SMRTbell Template Prep Kit 1.0; PacBio with barcoded SMRTbell adapters). SMRT sequencing was performed on the Sequel platform with Sequel v3.0 chemistry (PacBio) with LR SMRT Cells with 2-h pre-extension and 20-h movies. Reads corresponding to each sample were extracted by their SMRTbell barcodes and circular consensus sequences were generated from the raw subreads using SMRT Link with a minimum number of pass set to 1. The program lima in the Iso-Seq3 pipeline (v3.1.2) was applied to remove the 5′ and 3′ dual barcodes and also obtain the unclustered FLNC reads. Parameters used were: lima --isoseq --dump-clips. We did not cluster the FLNC reads further because highly identical paralogous transcripts could undesirably cluster together in this step. Due to the variability of the yields of FLNC transcripts across samples and loci, we used data from various combinations of samples in subsequent analyses when applicable.

To determine the coding potential for each FLNC transcript, we use the program ANGEL (v3.0) to call open reading frames with a minimum length cutoff of 200 amino acids and proper stop codons. All transcripts with open reading frames were mapped to the seven *TCAF* haplogroups (Supplementary Data 4 and 5) using minimap2 (v2.17-r941) and individual best placements to specific haplogroups were determined using sequence identity. Given the recent history of the *TCAF* duplications, we required a >99% sequence identity for an alignment to be considered a match between an FLNC read and the haplogroup contig and most have at least an overlap of 200 bases. To improve the sensitivity of FLNC read assignments, we allowed multi-mapping placements for an FLNC read as long as the difference in sequence identity between alignments is <0.01%. Finally, to determine gene models and isoforms in each haplogroup, we focused our analysis on FLNC transcripts with more than 10 reads (Supplementary Data 5).

**SNV calling for unique diploid sequences at the *TCAF* locus**. To perform population genetic analyses, we generated a joint call set for 840 published high-coverage short-read Illumina genomes, including 828 Human Genome Diversity Project (HGDP) samples, eight chimpanzee samples, and the four high-coverage archaic genomes[17–22]. Because the current human reference GRCh38 has a gap at the *TCAF* locus, we constructed two custom references for chromosome 7: one with Haplogroup 4 and the other with Haplogroup 5, in addition to a *TCAF*-SDs-hardmasked (chr7:143521769–143874696, GRCh38) chromosome 7. Paired-end data were aligned to the two custom references using BWA-MEM (v0.7.12), while ancient DNA data were mapped using BWA (v0.7.17) with parameters: −n 0.01 −o 2 −l 16500[19]. SNV genotypes were jointly generated using haplotype caller Free-Bayes (v1.0.2-6-g3ce827d) with the following parameters: "--min-coverage 10 --use-best-n-alleles 4". To ensure genotype quality, we excluded variants that were found within 10 bp of indels and have a quality score (QUAL) < 20. We identified a total of 1275 and 1295 SNVs from the call sets of Haplogroups 4 and 5, respectively, among the 840 samples. Note that we excluded 38 and 30 samples from the call sets of Haplogroups 4 and 5, respectively, due to >10% missing data to ensure the quality of genotype calls and downstream analyses. Because the results and interpretations of all downstream analyses are highly compatible, we concluded no obvious reference biases between the two sets. Unless otherwise stated, we reported results from the Haplogroup 4 call set because it has slightly better mapping quality.

**Phylogenetic analyses**. To infer the phylogenetic relationships for *TCAF* SDs and the unique diploid sequences in primates, we performed both maximum likelihood (IQ-TREE, v1.6.11)[56] and Bayesian phylogenetic-based (BEAST v2.5.0)[57] analyses. First, we determined parameters and models using ModelGenerator (v0.85)[58] and compared with the recommendations from the model-selection feature of IQ-TREE (using the parameters: "-bb 1000 -redo -nt 1 -m MFPMERGE"). Based on the best-fit model recommended, to run BEAST, in general, we set GAMMA Category Count = 5, shape parameter = 0.1, Proportion Invariant = 0.38, and using the GTR substitution model (all rates = 1.0) for the Site Model. For tree priors, we used Calibrated Yule Model, and while kept most of the parameters of the priors as default, for birthRate and clockRate we used Gamma (0.001, 1000) and set calibrations based on human–chimpanzee and human–gorilla divergence following the distributions of a log-normal ($M = 6,500,000$, $S = 0.09$) and log-normal($M = 10,500,000$, $S = 0.09$), respectively. For the analysis involving the inferred archaic hominin haplotypes, we also set dates for terminal samples to account for the differences between contemporary and fossil samples. For each run, we drew a date (in years) from a uniform [36,000, 100,000] for the Neanderthals and a uniform [50,000, 80,000]. To test if the assumption for a strict mutation clock rate of sequence evolution on all lineages is violated, we ran two replicates for the Clock Model with the Strict Clock model and the model of Relaxed Clock Log Normal[59] over branches on the tree separately. Overall, we find that both the distributions of overall tree heights and likelihoods are highly compatible between the two models, and that the posterior density for the long-normal standard deviation includes zero (Supplementary Fig. 43), suggesting that the Strict Clock model cannot be rejected. Thus, we chose to present inference results based on the Strict Clock model throughout this work. For each locus, we performed five independent runs to infer the phylogeny using a chain length of 50,000,000 samples and recorded every 1000 samples. We used the accompany program Tracer (v.1.7.1) to determine the quality of each run and, in general, we used the first 10% as burn-in and only kept runs with ESS > 500. All phylogenetic trees were plotted using Figtree (v1.4.3). We note that the 95% HPD intervals for the human–chimpanzee and human–gorilla divergences are [$5.9 \times 10^6$, $7.8 \times 10^6$] and [$8.6 \times 10^6$, $1.1 \times 10^7$] years ago, respectively, and are compatible to other published estimates[60]. However, the 95% highest-posterior density (HPD) of the strict clock rate is [$6.2 \times 10^{-10}$, $9.9 \times 10^{-10}$] per substitution per year, which is higher than the recent pedigree-based mutation rate of $4.3 \times 10^{-10}$ per nucleotide per year (95% confidence interval = $3.7 \times 10^{-10} - 4.7 \times 10^{-10}$)[61,62].

In addition, to validate the BEAST inferred tree, we reconstructed the phylogeny for the same sequences using both maximum parsimony and neighbor-joining[63] methods, implemented in MEGA X (v10.1.8)[29]. We used the default settings and ran each analysis with 1,000 bootstrap samples. Although bootstrap support for both trees was low overall, we find that the maximum-parsimony tree closely recapitulates the topology of the BEAST tree (Supplementary Fig. 44). While the neighbor-joining tree differs slightly in its topology, both trees suggest that the archaic hominin haplotypes are more closely related to human supergroup H4 than H5 and thus unlikely to be derived from ancestral ape sequence. These findings are consistent with the BEAST results.

**Population genetic analyses**. All population genetic analyses in this study were based on SNVs from the three unique diploid regions at the *TCAF* locus (chr7:143,501,000–143,521,000, chr7:143,729,525–143,741,525, chr7:143,875,000–143,895,000 [GRCh38]) using the HGDP, archaic hominin, and chimpanzee samples. Haplotypes were inferred by applying BEAGLE (v5.1-25Nov19.28d) with default settings to modern human, archaic hominin, and chimpanzee samples separately. We explored the diversity of the *TCAF* haplotype using a haplotype-based PCA to classify and assign haplotypes into individual clusters. Cluster assignments were based on the supervised k-means algorithm

using the top N informative PCs, where $\sum_{i=1}^{N} Var(PC_i) > 0.9$. To determine the best number of cluster k, for each possible k between 1 and 20, we computed the within groups sum of squares (WSS) using a distance matrix among the top N PCs and determined the best k when $WSS_{k+1} - WSS_k < delta$, where delta was chosen as 500 according to our analysis. Data visualization of the haplotype clustering was performed using the machine-learning technique t-distributed stochastic neighbor embedding (t-SNE)[28] implemented in the R package Rtsne (v0.15) using the following parameters: "perplexity = 50, max_iter = 5000, early_exag_coeff = 12, exaggeration = 4, stop_lying_iter = 1000, check_duplicates = FALSE, dims = 3". We optimized the t-SNE analysis by running five replicates of this computation and identifying the smallest Kullback-Leibler distance as the candidate for final visualization[28].

To test for signals of natural selection, we computed the Tajima's $D$[64] and Fay and Wu's $H$ statistics[30] over 2,000 bp windows over the 52.3 kbp unique diploid sequences for individual populations as well as the archaic samples. Because these statistics are sensitive to population history, we performed a large coalescent simulation using both MaCS[65] and msprime (v0.7.4)[66] and demographic models as described previously[25,67] that were implemented in the Python package Stdpopsim (v0.1.2)[68]. For each of the models, we generate 10,000 whole-genome simulations to account for past histories between modern and archaic hominins and among multiple African and non-African populations (Supplementary Data 9) and use chimpanzee[27] as the outgroup. Note that our simulations carefully match the local mutation rate and recombination rate variation at these regions to avoid possible biases to our selection inferences. To estimate the TMRCA for a locus of interest, we used the Thomson's estimator[69]. To perform a haplotype-based PCA, we first recode haplotype bases as numeric codes, where 0 and 1 (and 2 if needed) are ancestral (using chimpanzee as the outgroup) and derived alleles, respectively. We performed the haplotype-based PCA following the suggestion described in Browning et al 2016[70] but using the R package irlba (v2.3.3) to calculate principal components. LD measurements among SNVs were computed using Lewontin's $D'$[71] and $R^2$ implemented in PLINK (v1.09)[54]. Note only SNVs with minor allele frequencies >10% were included for this analysis because lower-frequency variants are less informative about linkage.

**Reporting summary**. Further information on research design is available in the Nature Research Reporting Summary linked to this article.

## Data availability

All data used in this study, including assembled BAC contigs (GenBank accession: MT985491 - MT985505 at https://www.ncbi.nlm.nih.gov/nuccore/) and Iso-Seq capture transcript data (BioProject: PRJNA657884). These data are available to anyone for purposes of reproducing or extending the analysis. In addition, the following publicly available data were downloaded: Human Genome Diversity Project (https://www.internationalgenome.org/data-portal/data-collection/hgdp), Simons Genome Diversity Project (https://docs.cancergenomicscloud.org/v1.0/docs/simons-genome-diversity-project-sgdp-dataset), Neandertal/Denisovan genomes (http://cdna.eva.mpg.de/neandertal/), Great ape nonhuman primate genomes (https://www.ncbi.nlm.nih.gov/bioproject/?term=PRJNA189439), GTEx multi-tissue data (release v8; https://gtexportal.org/home/), Human Protein Atlas (HPA; https://www.proteinatlas.org).

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

## Acknowledgements

The authors thank T. Brown for assistance in editing this manuscript. We also thank S.C. Murali and D.S. Gordon for help submitting data to the NCBI database. Funding: This work was supported, in part, by the US National Institutes of Health (NIH) grant R01HG002385 to E.E.E. P.H. is supported by the NIH Pathway to Independence Award (NHGRI, K99HG011041). S.C. was supported by a National Health and Medical Research Council (NHMRC) C. J. Martin Biomedical Fellowship (1073726). E.E.E. is an investigator of the Howard Hughes Medical Institute.

## Author contributions

P.H. and E.E.E. designed and planned experiments. V.D., C.B., S.C., A.P.L., K.M.M., M.S., A.E.W. and J.G.U. prepared libraries and generated and analyzed sequencing data. P.H., V.D., M.R.V. and T.H. performed variant calling and bioinformatics analyses. P.H., V.D., M.R.V. and P.C.D. analyzed long-read sequencing data and assembled contigs. P.H. and Y.M. performed phylogenetic inferences. P.H. performed population genetic analyses. A.P.L., K.M.M., P.C.D. and J.G.U. generated Iso-Seq transcript data. P.H. and E.E.E. wrote the manuscript.

## Competing interests

J.G.U. is an employee of Pacific Biosciences, Inc. The rest of the authors declare no competing interests.
