## [Peer Review File · Nature Communications]

Evidence for opposing selective forces operating on human-specific duplicated TCAF genes in Neanderthals and humansREVIEWER COMMENTS

Reviewer #1 (Remarks to the Author):

Hsieh et al. analyzed a locus with complex duplication history containing the TCAF genes that may play a role in the sensation of cold. With additional long-read sequencing data the authors were able to reconstruct the haplotypes present in humans and apes. Analysis of these haplotypes and comparisons to archaic humans suggest that TCAF duplications formed more than 1myrs ago and were later lost in archaic humans. Based on comparisons of Tajima's D to simulations, the authors suggest that the locus evolved under positive selection in archaic humans and diversifying selection in Native Americans, Melanesian and Siberian populations.

The authors performed a detailed study on a highly complex and challenging genomic locus containing genes that have an interesting function. The results are clear and well described. I do not doubt that this work will appeal to a wide range of readers with interest in human evolution.

I have two main comments for the authors:

1) The presence of a haplotype in Neandertals and Denisovans that resembles that of the apes, in structure, would typically be interpreted as a sign for a modern human specific duplication. I understand that the authors exclude this scenario based on a BEAST analysis that suggests that the archaic humans carry a haplotype that is more closely related to Haplogroup 4, 3(-1&-2) and 1. Given that mutation rates may increase following a duplication, can you think of any scenario that would place the archaic haplotype ancestral to the expansion in modern humans? Can you exclude such a scenario based on a parimony tree or a neighbor joining tree that is not explicitly relying on assumptions of similar rates of sequence evolution on all lineages?

2) I am highly sceptical of claims of selection based on simulations alone. Even though the authors sample from within the range of published estimates for various parameters for their simulations, these estimates are always a simplification. For instance, in Hsieh et al. 2019, an N_e of 2000-4000 for Neandertals, Denisovans and their ancestor is assumed. However, according to the PSMC estimates of effective population sizes, we cannot exclude the possibility that N_e may have been substantially higher in the ancestor. I find it particularly problematic that the authors claim evidence of selection when the distribution of the simulations is shifted compared to the analysis of sliding windows (if I understand correctly): The random simulations for the archaics did produce a number of Tajima's D values that seem to overlap the observed range (Fig. 5C, lower panel). Can the authors show that the observed Tajima's D for this locus is exceptional compared to other loci in the genome to further support the claim of positive selection? I have similar concerns regarding the diversifying selection in some populations.

Minor:

- Please state the mutation rate that you assumed for estimating phylogeny's. The estimate of ~6 million years for the human-chimpanzee divergence appears very young for the typically assumed mutation rate of $\sim 0.5 \times 10^{-8}$. Similarly, a divergence of 0.5Myr is just barely compatible with the archaic-human split time estimates and would mean that the locus diverged very close to that time.

- line 158: please introduce DupA, DupC in the text in addition to the figures.

- line 56: "stayed as" could be "remained"

Reviewer #2 (Remarks to the Author):

Hsieh et al., present a comprehensive and detailed study of the structure and history of complex genomic region encoding for TRP channel-associated factor 1/2 (TCAF1/TCAF2) proteins. The study involves huge amounts of work. The authors established the CN in multiple human populations, archaic hominins and some primates. They also reconstructed the history of duplications. This work substantially improves our knowledge of this genomic region and the two TCAF genes. Unfortunately the evidence of selection and functional consequences of the duplications is weaker. I agree with the authors that duplications are a source of innovation but the manuscript does not establish that this is the case for TCAF1/TCAF2.

The study of the structure and history of structural changes is remarkable. This is not my area of expertise but it is the expertise of this group so I have no doubt that this section is right. It is also necessary to study the evolution and function of this genomic region.

The evidence of selection needs to be stronger if this is a central message. The effective population size of archaic hominins was small so low levels of diversity are not surprising. I am not convinced that Tajima's D can be used as evidence of purifying selection. These few genomes do not represent a population so the expected Tajima's D distribution is unknown. The authors used simulations but we

know little about the demography of these populations and demographic models are rough approximations. This limitation should be clearly acknowledged. To strengthen the evidence of selection the manuscript should show that the simulations recapitulate the genome-wide distribution of Tajima's D values in 2,000bp segments. If that is the case we can interpret low Pvalues as strong evidence of selection. Otherwise the evidence is weak. It would also be good to add an explanation for why negative Tajima's D values are interpreted as positive selection rather than strong purifying selection.

The evidence of selection in humans is also weak. Tajima's D Pvalues are based on simulations. Figure 5 legend and Methods state that the simulations are based on 1,000 different demographic models. Why 1,000 models? What models, since the reference is to one? How well do they recapitulate the Tajima's D values in 2,000bp segments in the genome? Several of the populations with significant Pvalues have complex demography so this is important. Another way to strengthen the evidence would be to show that the structure of adjacent 2,000bp segments have Tajima's D values that agree with expectations under natural selection.

The results for functional effects of changes in the region are interesting but also not definite. The functional relevance of the eQTLs is ambiguous. If the functional relevance of TCAF1 and TCAF2 is by regulating TRPM8, are the tissues where eQTLs are identified relevant in terms of TRPM8? Does TRPM8 express at high levels in these tissues? Likewise, the associations with disease are only nominal and in single populations. More evidence is needed to show that they are biologically relevant.

The correlation of CN with latitude is interesting. Exploring this in more detail would strengthen the paper because the authors propose temperature and diet as selection forces. The authors could for example ask if the correlation is higher than observed in other segments of the genome.

Minor points.

The color scheme in Figure 1C could be better. It is hard to see differences among few copy numbers with small figures and similar colors.

The estimate for the age of DupB is older than estimates of the age of a single copy region. Is this meaningful or informative? Is that single copy region representative of our expectations in the genome? Why in p.244 the reference for this single copy locus is to Figure 2?

The authors would help the reader if they explained haplotype-based PCA. In Figure 5A the main figure and the zoom figure look different. Why is that?

L 382. Is the importance of TCAF in thermal regulation and sensing established? If it is, explaining how that works would strengthen the paper and make the interpretation less dependent on TRPM8 and its expression.

Responses to REVIEWER COMMENTS

Reviewer #1 (Remarks to the Author):

Hsieh et al. analyzed a locus with complex duplication history containing the *TCAF* genes that may play a role in the sensation of cold. With additional long-read sequencing data the authors were able to reconstruct the haplotypes present in humans and apes. Analysis of these haplotypes and comparisons to archaic humans suggest that *TCAF* duplications formed more than 1myrs ago and were later lost in archaic humans. Based on comparisons of Tajima's D to simulations, the authors suggest that the locus evolved under positive selection in archaic humans and diversifying selection in Native Americans, Melanesian and Siberian populations.

The authors performed a detailed study on a highly complex and challenging genomic locus containing genes that have an interesting function. The results are clear and well described. I do not doubt that this work will appeal to a wide range of readers with interest in human evolution.

I have two main comments for the authors:

1) The presence of a haplotype in Neandertals and Denisovans that resembles that of the apes, in structure, would typically be interpreted as a sign for a modern human specific duplication. I understand that the authors exclude this scenario based on a BEAST analysis that suggests that the archaic humans carry a haplotype that is more closely related to Haplogroup 4, 3(-1&-2) and 1. Given that mutation rates may increase following a duplication, can you think of any scenario that would place the archaic haplotype ancestral to the expansion in modern humans? Can you exclude such a scenario based on a parimony tree or a neighbor joining tree that is not explicitly relying on assumptions of similar rates of sequence evolution on all lineages?

The presence of high-identity duplications in the ancestral haplogroup 4 indeed would have the potential to undergo non-allelic homologous recombination at a much higher rate creating the potential for reversion back to the ancestral structure in apes through deletion. The alternative scenario would be incomplete lineage sorting retaining the ancestral configuration. The latter, however, is not supported by phylogenetic analyses. However, the reviewer is correct that we inferred this (**Fig. 4b**, the BEAST analysis) under the assumption of a strict molecular clock. We further explored this by rerunning the analysis with a relaxed clock model under a log-normal distribution (Drummond et al. 2006) and repeated this analysis with a replicate (**Supplementary Fig. 40**). Overall, both the relaxed and strict model gave similar results. The distributions of overall tree heights and likelihoods are highly compatible. Notably, the posterior density for the log-normal standard deviation includes zero, suggesting that the strict clock model cannot be rejected in favor of the relaxed clock model. Thus, our inferred phylogeny is robust under these two conditions.

Supplementary Fig. 40. Comparisons of key statistics between the models of an uncorrelated log-normal relaxed clock and a strict clock. For each model, two replicates were performed (run 1 and run 2). (A) Density of likelihood, (B) density of tree height, and (C) density of standard deviation for clock rate under an uncorrelated log-normal relaxed clock model. Note that the distributions of the standard deviation of clock rates under the relaxed clock model overlap with zero (i.e., no rate variation).

Also, as suggested by the referee, we reconstructed the phylogeny for the same 12.3 kbp single-copy unique sequences using both maximum parsimony (**Supplementary Fig. 41a**; Nei and Kumar 2000) and neighbor-joining (**Supplementary Fig. 41b**; Saitou and Nei 1987) methods, implemented in MEGA X (Stecher et al. 2020). We used the default settings and ran each analysis with 1000 bootstrap samples. Although the bootstrap support for both trees was low overall, we find that the maximum-parsimony tree closely recapitulates the topology of the BEAST tree (**Fig. 4b**). While the neighbor-joining tree differs slightly in its topology, all three trees suggest that the archaic hominin haplotypes are more closely related with supergroup H4 than H5 and, thus, unlikely to be derived from ancestral ape sequence. Hence, these findings are consistent with the BEAST results.

Given that the archaic and modern human haplotypes are nested within the inferred phylogeny, the structural rearrangement of the *TCAF* locus started >1.7 Mya, and the estimated TMRCA for all hominin haplotypes is ~500-700 kya, we parsimoniously conclude that despite the observed archaic haplotypes resembling the structure of nonhuman primates, they emerged within the ancestral humans prior to the divergence between archaic and modern humans and, thus, are not ancestral to modern human haplotypes.

In our revision, we clarified our methods and added these new results to the RESULTS and METHODS sections.

(RESULTS) Lines 258-262: “We should note that these time estimates are subject to uncertainties, such as mutation rate, and therefore caution is warranted when interpreting these dates (**Methods**). Nevertheless, multiple phylogenetic methods (**Fig. 4b and Supplementary**

Fig. 22) show that all archaic haplotypes and the supergroup H4 are more closely related than either is to the supergroup H5, and the clade of archaic and supergroup H4 haplotypes coalesces ~0.53 Mya (**Fig. 4b**).”

(METHODS) Lines 581-587: “To test if the assumption for a strict mutation clock rate of sequence evolution on all lineages is violated, we ran two replicates for the Clock Model with the Strict Clock model and the model of Relaxed Clock Log Normal⁵⁷ over branches on the tree separately. Overall, we find that both the distributions of overall tree heights and likelihoods are highly compatible between the two models, and that the posterior density for the long-normal standard deviation includes zero (**Supplementary Fig. 40**), suggesting that the Strict Clock model cannot be rejected. Thus, we chose to present inference results based on the Strict Clock model throughout this work.”

(METHODS) Lines 596-603: “In addition, to validate the BEAST inferred tree, we reconstructed the phylogeny for the same sequences using both maximum parsimony and neighbor-joining⁶¹ methods, implemented in MEGA X²⁹. We used the default settings and ran each analysis with 1000 bootstrap samples. Although bootstrap support for both trees was low overall, we find that the maximum-parsimony tree closely recapitulates the topology of the BEAST tree (**Supplementary Fig. 41**). While the neighbor-joining tree differs slightly in its topology, both trees suggest that the archaic hominin haplotypes are more closely related to human supergroup H4 than H5 and thus unlikely to be derived from ancestral ape sequence. These findings are consistent with the BEAST results.”

Supplementary Fig. 33. Inferred phylogenies of the modern human and archaic hominin haplotypes using the 12 kbp unique sequences embedded within *TCAF* SDs. Phylogenies were inferred using (A) maximum parsimony and (B) neighbor-joining methods, implemented in MEGA X (v11.0). Haplotypes are the same used in Figure 4B from the main text. Branches with posterior probabilities <90% are colored in red. The branch length of the maximum parsimony tree refers to the number of substitutions per unit length, while that of the neighbor-joining tree is the number of substitutions per site for a given length unit.

Fig. 4b (from the main text). Inferred phylogeny of the modern human haplogroups and archaic hominin haplotypes using the 12 kbp unique sequences embedded within TCAF SDs (Fig. 1). Haplotypes of archaic samples were generated using high-confident single-nucleotide variants (SNVs) called within the unique diploid region. Phylogenetic inference was performed using BEAST (v2.5.0). Branches with posterior probabilities <90% are colored in red.

Reference:

1. Drummond, A. J., Ho, S. Y., Phillips, M. J., & Rambaut, A. (2006). Relaxed phylogenetics and dating with confidence. *PLoS Biology*, 4, e88.
2. Nei M. and Kumar S. (2000). *Molecular Evolution and Phylogenetics*. Oxford University Press, New York.
3. Saitou N. and Nei M. (1987). The neighbor-joining method: A new method for reconstructing phylogenetic trees. *Molecular Biology and Evolution* 4:406-425.
4. Stecher G., Tamura K., and Kumar S. (2020). *Molecular Evolutionary Genetics Analysis (MEGA) for macOS*. *Molecular Biology and Evolution* (<https://doi.org/10.1093/molbev/msz312>).

2) I am highly sceptical of claims of selection based on simulations alone. Even though the authors sample from within the range of published estimates for various parameters for their simulations, these estimates are always a simplification. For instance, in Hsieh et al. 2019, an N_e of 2000-4000 for Neandertals, Denisovans and their ancestor is assumed. However, according to the PSMC estimates of effective population sizes, we cannot exclude the possibility that N_e may have been substantially higher in the ancestor. I find it particularly problematic that the authors claim evidence of selection when the distribution of the simulations is shifted compared to the analysis of sliding windows (if I understand correctly): The random simulations for the archaics did produce a number of Tajima's D values that seem to overlap the observed

range (Fig. 5C, lower panel). Can the authors show that the observed Tajima's *D* for this locus is exceptional compared to other loci in the genome to further support the claim of positive selection? I have similar concerns regarding the diversifying selection in some populations.

The referee is correct that simulations are only approximations and as good as the underlying assumptions. Based on the referee's suggestion, we compared the observed Tajima's *D* value for *TCAF* to that of the genome by computing genome-wide empirical Tajima's *D* distributions for using the same 2,000 bp windows for individual archaic and modern human groups (Fig. 5c). Consistent with our original results, we find that the Tajima's *D* distribution for the *TCAF* locus from archaic groups (as well as Neanderthal samples without the Denisovan) is significantly negative when compared to the rest of the genome (Bonferroni's *p* values < 5.1e-5). In contrast, the *TCAF* Tajima's *D* distributions among most modern human groups are compatible with those of the genome. Two exceptions are the Native American and Melanesian groups which show significantly positive *D* values at the *TCAF* locus when compared to its empirical genome-wide distribution (Fig. 5c).

Fig. 5c. Comparisons of the Tajima's *D* distributions between the *TCAF* locus and the entire genome in human and archaic populations. Tajima's *D* statistics are computed for individual archaic and modern groups across the entire genome (grey) versus *TCAF* (red) based on 2,000 bp windows. *P* values are empirically determined by testing if the observed values at *TCAF* are significantly negative (green) than those from the genome, except for the American and Melanesian groups, where their *p* values represent if the observed *D* values at

TCAF are significantly positive (*italic*) than the those from the genome. Bold fonts indicate Bonferroni's p values < 0.05.

While genome-wide outliers of the *D* distribution represent good candidates of selection targets, locally deviated *D* values at the *TCAF* locus from the expectation under reasonable demographic models could also suggest possible actions of selection. To this end, we further explored possible impacts on our selection inference results due to uncertainty of the ancestral population size among the archaic hominins. We simulated a model published recently by Jacobs et al. (2019). This model incorporates a larger ancestral size of the Neanderthal and Denisovan groups ($N_e=13,239$) than the 2000 to 4000 size used in our original simulations. Simulations were generated based on the implementation from the PopSim consortium – an open-source population genetics project (StdpopSim; Adrion et al. 2020). In addition to models used in our original submission (Hsieh et al. 2019), we performed new simulations using the following published models for individual modern human population groups to explore possible biases due to misspecification of underlying demographic models (**Table S9**). For each model, we generated 10,000 whole-genome replications and computed Tajima's *D* values using the same 2,000 bp windows.

Supplementary Table 9. Summary of additional models used for coalescent simulations.

Super population	Reference	Model ID in StdpopSim
African	Jouganous et al. 2017	OutOfAfrica_4J17
American	Browning et al. 2011	AmericanAdmixture_4B11
East Asian	Jouganous et al. 2017	OutOfAfrica_4J17
European	Jouganous et al. 2017	OutOfAfrica_4J17
Middle Easterner	Gladstein and Hammer 2019	AshkSub_7G19
Melanesian	Malaspinas et al. 2016	Melanesian_6H19
Siberian	Hsieh et al. 2017	-
South Asian	Terhorst et al. 2019	-
Archaic hominin	Jacobs et al. 2019	PapuansOutOfAfrica_10J19

To summarize the overall performance of these models, we consider comparisons between the observed and simulated genome-wide statistics. We first would like to note that the main model used for our original submission (Hsieh et al. 2019) did show a significant correlation between observed and simulated data using Watterson's theta statistic (**Response to Review Figure 1; copied from Hsieh et al. 2019, Fig. S9**). However, we recognized that this model was inferred

specifically for Africans, East Asians, and Melanesians and was derived from different population samples than those used in the current study.

Response to Review Figure 1. (Copied from Hsieh et al. 2019, Figure S9) Whole-genome simulations accurately capture the local genetic diversity in the real data. Estimates of nucleotide diversity (Watterson's theta) were computed using windows of 100 SNVs for both the real and simulated data. Pearson's correlation coefficient test was used to assess the agreement between real and simulated whole-genome data.

Overall, the simulated Tajima's D distributions based on different coalescent models qualitatively recapitulate the observed data (**Supplementary Fig. 32**); however, in all cases we note that there are clear differences across all populations. These differences are likely, at least in part, because each of our super-population groups includes multiple geographically closed populations in the Human Genome Diversity Project, and, also in part, because these models are inferred using different population samples in individual studies (e.g., the 1000 Genomes Project and Simons Genome Diversity Project).

Supplementary Fig. 32. Comparisons of genome-wide Tajima's D statistic values between observed and simulated data using models listed in Supplementary Table 9. Tajima's D values are computed using the same 2,000 bp windows across the genome as described in the main text.

To determine if our original observations of selection at the *TCAF* locus still hold under these new models, we compared the observed Tajima's D values to those computed from these new simulations. Consistent with our original findings, in both archaic groups, we observed significantly negative Tajima's D values from the data compared with those from the simulations (**Supplementary Fig. 33**). Within modern humans, no populations show significantly negative Tajima's D values compared with simulated data; however, we note that the American population samples do have significantly larger Tajima's D values than those based on simulations (**Supplementary Fig. 33**). In addition, we notice that inconsistent with our original finding, the *TCAF* Tajima's D values of Melanesians becomes insignificant after accounting for multiple testing.

Supplementary Figure 33. Comparisons of Tajima's D statistic values between observed and simulated $TCAF$ data using models listed in Supplementary Table 9. Tajima's D values were computed using 2,000 bp windows as described in the main text. P values were computed by testing if the observed values are significantly negative than those from simulated data, except for the American and Melanesian groups, where their p values represent if the observed Tajima's D values are significantly positive (*italic*) than those from the simulated data. Bold fonts indicate Bonferroni's p values < 0.05.

In summary, application of these different models recapitulates two of our original observations. We find that $TCAF$ Tajima's D values in the two archaic and Native American groups are 1) significantly different from those of the genome, and 2) significantly negative and positive, respectively, compared to their simulated $TCAF$ values. Thus, our new data provide additional evidence for and are consistent with our original findings of opposing selection forces in archaic and modern hominins.

Given these new results, we now revise the RESULTS and DISCUSSION sections of our manuscript as follows.

(RESULTS) Lines 290-313: "To formally test whether natural selection plays a role in these unusual patterns in both modern and archaic humans, we performed a selection test using the Tajima's D statistic and coalescent simulations under plausible demographic models to assess significance (**Methods**). In several human groups, we noticed that the Tajima's D values at

TCAF significantly deviate from those across the whole genome (**Fig. 5c**). Specifically, the Tajima's *D* values of the archaic hominins are significantly negative compared to the rest of the genome (median Tajima's *D* < -0.99; Bonferroni's *p* values $\leq 4.15 \times 10^{-7}$), regardless of the presence of the Denisovan individual in the test. In contrast, while the *TCAF* Tajima's *D* distributions among most modern human groups are compatible with those of the genome, there are two notable exceptions: Native American (median Tajima's *D* = 1.62; Bonferroni's *p* = 5.1×10^{-5}) and Melanesian (median Tajima's *D* = 0.69, Bonferroni's *p* = 0.01406) samples (**Fig. 5c**).

To account for biases in the Tajima's *D* values due to demographic effects, we simulated data using multiple population-specific models published recently (**Supplementary Table 9; Methods**) and computed Tajima's *D* values across 2,000 bp windows. Overall, the simulated Tajima's *D* distributions qualitatively recapitulate the observed data; however, in all cases we note differences in individual populations (**Supplementary Fig. 32**). These differences are likely due, in part, to the presence of multiple geographically closed populations in each supergroup, and, also in part, due to different population samples underlying the demographic models used in our study. Consistent with the results of the comparison between the *TCAF* locus and the whole genome, in both archaic hominin groups, we observe significantly negative Tajima's *D* values from the data compared with those from the simulations (Bonferroni's *p* values $\leq 6.62 \times 10^{-6}$; **Supplementary Fig. 33**). Within modern humans, no populations show significantly negative Tajima's *D* values compared with simulated data though we do note that the Native American samples show significantly more positive Tajima's *D* values than those based on simulations (Bonferroni's *p* value = 3.32×10^{-11} ; **Supplementary Fig. 33**), possibly as a result of balancing selection at this locus in this population."

(DISCUSSION) Lines 433-442: "In contrast, Neanderthal and Denisovan show a paucity of genetic variation, and while the sample size is still limited, this observation is unlikely to change with the sequencing of additional archaic genomes. The strongly negative Tajima's *D* and Fay and Wu's *H* values at this locus compared with those of the genome-wide and simulated values, which accounts for differences explained by demographic history, suggest that the observation of the single-haplotype without duplication in the archaic samples is highly unlikely under neutrality. We note that our selection inference is limited by our understanding regarding the demography of these archaic hominins and the underlying models and should be further tested in the future by incorporating new archaic samples and demographic models. Nevertheless, given our inference results, we argue that positive selection has reduced genetic diversity at the *TCAF* locus in these archaic hominin lineages."

References:

- 1) Hsieh P, Vollger MR, Dang V, Porubsky D, Baker C, Cantsilieris S, Hoekzema K, Lewis AP, Munson KM, Sorensen M, Kronenberg ZN. Adaptive archaic introgression of copy number variants and the discovery of previously unknown human genes. *Science*. 2019 Oct 18;366(6463).

- 2) Adrion JR, Cole CB, Dukler N, Galloway JG, Gladstein AL, Gower G, Kyriazis CC, Ragsdale AP, Tsambos G, Baumdicker F, Carlson J. A community-maintained standard library of population genetic models. *Elife*. 2020 Jun 23;9:e54967.
- 3) Jouganous J, Long W, Ragsdale AP, Gravel S. Inferring the joint demographic history of multiple populations: beyond the diffusion approximation. *Genetics*. 2017 Jul 1;206(3):1549-67.
- 4) Browning SR, Browning BL, Daviglus ML, Durazo-Arvizu RA, Schneiderman N, Kaplan RC, Laurie CC. Ancestry-specific recent effective population size in the Americas. *PLoS genetics*. 2018 May 24;14(5):e1007385.
- 5) Gladstein AL, Hammer MF. Substructured population growth in the ashkenazi jews inferred with approximate bayesian computation. *Molecular biology and evolution*. 2019 Jun 1;36(6):1162-71.
- 6) Malaspinas AS, Westaway MC, Muller C, Sousa VC, Lao O, Alves I, Bergström A, Athanasiadis G, Cheng JY, Crawford JE, Heupink TH. A genomic history of Aboriginal Australia. *Nature*. 2016 Oct;538(7624):207-14.
- 7) Hsieh P, Hallmark B, Watkins J, Karafet TM, Osipova LP, Gutenkunst RN, Hammer MF. Exome sequencing provides evidence of polygenic adaptation to a fat-rich animal diet in indigenous Siberian populations. *Molecular biology and evolution*. 2017 Nov 1;34(11):2913-26.
- 8) Terhorst J, Kamm JA, Song YS. Robust and scalable inference of population history from hundreds of unphased whole genomes. *Nature genetics*. 2017 Feb;49(2):303-9.
- 9) Jacobs GS, Hudjashov G, Saag L, Kusuma P, Darusallam CC, Lawson DJ, Mondal M, Pagan L, Ricaut FX, Stoneking M, Metspalu M. Multiple deeply divergent Denisovan ancestries in Papuans. *Cell*. 2019 May 2;177(4):1010-21.

Minor:

- Please state the mutation rate that you assumed for estimating phylogeny's. The estimate of ~6 million years for the human-chimpanzee divergence appears very young for the typically assumed mutation rate of $\sim 0.5 \times 10^{-8}$. Similarly, a divergence of 0.5Myr is just barely compatible with the archaic-human split time estimates and would mean that the locus diverged very close to that time.

For our phylogenetic inference, we estimated the 95% highest-posterior density (HPD) of the strict clock rate is $[6.2 \times 10^{-10}, 9.9 \times 10^{-10}]$ per substitution per year. The 95% HPD intervals for the human–chimpanzee and human–gorilla divergences are $[5.9 \times 10^6, 7.8 \times 10^6]$ and $[8.6 \times 10^6, 1.1 \times 10^7]$ years ago, respectively. While we noticed that our estimated mutation rate is higher than the pedigree-based mutation rate of 4.3×10^{-10} per nucleotide per year (95% confidence

interval= 3.7×10^{-10} – 4.7×10^{-10}) (Rahbari et al. 2015, Sasani et al. 2019), our estimated divergence times overlap with other published estimates (Langergraber et al. 2012).

We now revise our METHODS and DISCUSSION sections to address these uncertainties in our inferences as follows.

(RESULTS) Lines 255-260: “Using these archaic hominin haplotypes with the sequences of the seven BAC haplogroups, we estimate the time to the most recent common ancestor (TMRCA) of all hominins being 0.77 Mya (95% C.I.:0.55-1.03 Mya), which is consistent with that reported above. We should note that these time estimates are subject to uncertainties, such as mutation rate, and therefore caution is warranted when interpreting these dates (**Methods**).”

(METHODS) Lines 591-596: “We note that the 95% HPD intervals for the human–chimpanzee and human–gorilla divergences are [5.9×10^6 , 7.8×10^6] and [8.6×10^6 , 1.1×10^7] years ago, respectively, and are compatible to other published estimates⁵⁸. However, the 95% highest-posterior density (HPD) of the strict clock rate is [6.2×10^{-10} , 9.9×10^{-10}] per substitution per year, which is higher than the recent pedigree-based mutation rate of 4.3×10^{-10} per nucleotide per year (95% confidence interval= 3.7×10^{-10} – 4.7×10^{-10})^{59,60}.”

References:

- 1) Langergraber KE, Prüfer K, Rowney C, Boesch C, Crockford C, Fawcett K, Inoue E, Inoue-Muruyama M, Mitani JC, Muller MN, Robbins MM. Generation times in wild chimpanzees and gorillas suggest earlier divergence times in great ape and human evolution. *Proceedings of the National Academy of Sciences*. 2012 Sep 25;109(39):15716-21.
- 2) Rahbari R, Wuster A, Lindsay SJ, Hardwick RJ, Alexandrov LB, Al Turki S, Dominiczak A, Morris A, Porteous D, Smith B, Stratton MR. Timing, rates and spectra of human germline mutation. *Nature genetics*. 2016 Feb;48(2):126-33.
- 3) Sasani TA, Pedersen BS, Gao Z, Baird L, Przeworski M, Jorde LB, Quinlan AR. Large, three-generation human families reveal post-zygotic mosaicism and variability in germline mutation accumulation. *Elife*. 2019 Sep 24;8:e46922.

- line 158: please introduce DupA, DupC in the text in addition to the figures.

We now revise and introduce DupA, DupB, and DupC in the main text as follows.

Lines 84-87: “We applied a read-depth-based genotyper¹⁶ to reevaluate copy number diversity of TCAF SDs (duplication segments A [DupA], B [DupB], and C [DupC]; **Fig. 1a**) in a collection of recently published diverse human and nonhuman samples¹⁷⁻²³.”

- line 56: "stayed as" could be "remained"

We revised according to the reviewers' suggestion.

Lines 54-55: “The TCAF family originated from an ancient gene duplication event at the basal of mammalian phylogeny and **remained** single-copy genes throughout much of their evolution.”

Reviewer #2 (Remarks to the Author):

Hsieh et al., present a comprehensive and detailed study of the structure and history of complex genomic region encoding for TRP channel-associated factor 1/2 (*TCAF1/TCAF2*) proteins. The study involves huge amounts of work. The authors established the CN in multiple human populations, archaic hominins and some primates. They also reconstructed the history of duplications. This work substantially improves our knowledge of this genomic region and the two *TCAF* genes. Unfortunately the evidence of selection and functional consequences of the duplications is weaker. I agree with the authors that duplications are a source of innovation but the manuscript does not establish that this is the case for *TCAF1/TCAF2*.

The study of the structure and history of structural changes is remarkable. This is not my area of expertise but it is the expertise of this group so I have no doubt that this section is right. It is also necessary to study the evolution and function of this genomic region.

The evidence of selection needs to be stronger if this is a central message. The effective population size of archaic hominins was small so low levels of diversity are not surprising. I am not convinced that Tajima's *D* can be used as evidence of purifying selection. These few genomes do not represent a population so the expected Tajima's *D* distribution is unknown. The authors used simulations but we know little about the demography of these populations and demographic models are rough approximations. This limitation should be clearly acknowledged. To strengthen the evidence of selection the manuscript should show that the simulations recapitulate the genome-wide distribution of Tajima's *D* values in 2,000bp segments. If that is the case we can interpret low *P* values as strong evidence of selection. Otherwise the evidence is weak.

There are two important central messages from this paper that we believe will be of broad interest to the genetics/evolutionary field. The first is the relatively rapid dynamic structural changes that can occur across a locus resulting in both human-specific changes and dramatic genomic differences between *Homo neanderthalensis/denisova* and *Homo sapiens*. The second is the rather surprising finding (for us) that there are apparent signatures of selection and that positive selection was pointing to Neanderthals as opposed to the modern humans where there are suggestions of diversifying selection. The referee, however, is correct in pointing out that our Tajima's *D* inference is limited by our understanding regarding the demography of these archaic hominins and the underlying models. As suggested, we have added a statement in the DISCUSSION section to clearly describe these limitations on our inference results for selection.

(DISCUSSION) Lines 435-442: "The strongly negative Tajima's *D* and Fay and Wu's *H* values at this locus compared with those of the genome-wide and simulated values, which accounts for differences explained by demographic history, suggest that the observation of the single-haplotype without duplication in the archaic samples is highly unlikely under neutrality. We note that our selection inference is limited by our understanding regarding the demography of these archaic hominins and the underlying models and should be further tested in the future by incorporating new archaic samples and demographic models. Nevertheless, given our inference results, we argue that positive selection has reduced genetic diversity at the *TCAF* locus in these archaic hominin lineages. "

In this revision, to strengthen our original findings for signals of positive selection in archaic hominins and diversifying/balancing selection in the Native American group, we now include additional analyses as follows:

- 1) a comparison between the local *TCAF* and genome-wide distributions of Tajima's *D* values,
- 2) new simulations based on additional, recently published models, including a model that considers a larger ancestral population size of the archaic hominins (**Supplementary Table 9**), and
- 3) application of the Fay and Wu's *H* test to distinguish positive selection from purifying selection on the condition of significantly negative Tajima's *D* values in the archaic groups.

We note that the first two analyses were also requested by Reviewer #1, and we have reproduced this analysis here for Reviewer #2.

First, we compared the observed Tajima *D* value for *TCAF* to that of the genome by computing genome-wide empirical Tajima's *D* distributions for using the same 2,000 bp windows for individual archaic and modern human groups (**Fig. 5c**). Consistent with our original results, we find that the Tajima's *D* distribution for the *TCAF* locus from archaic groups (as well as Neanderthal samples without the Denisovan) is significantly negative when compared to the rest of the genome (Bonferroni's *p* values < 1e-6). In contrast, the *TCAF* Tajima's *D* distributions among most modern human groups are compatible with those of the genome. Two exceptions are the Native American and Melanesian groups which show significantly positive *D* values at the *TCAF* locus when compared to its empirical genome-wide distribution (**Fig. 5c**).

Supplementary Fig. 30. Comparisons of the Tajima's *D* distributions between the *TCAF* locus and the entire genome in human and archaic populations. Tajima's *D* statistics are

computed for individual archaic and modern groups across the entire genome (grey) versus *TCAF* (red) based on 2,000 bp windows. P values are empirically determined by testing if the observed values at *TCAF* are significantly negative (green) than those from the genome, except for the American and Melanesian groups, where their p values represent if the observed *D* values at *TCAF* are significantly positive (italic) than the those from the genome. Bold fonts indicate Bonferroni's p values < 0.05.

While genome-wide outliers of the *D* distribution represent good candidates of selection targets, locally deviated *D* values at the *TCAF* locus from the expectation under reasonable demographic models could also suggest possible actions of selection. To this end, we further explored possible impacts on our selection inference results due to uncertainty of the ancestral population size among the archaic hominins. We simulated a model published recently by Jacobs et al. (2019). This model incorporates a larger ancestral size of the Neanderthal and Denisovan groups ($N_e=13,239$) than the 2000 to 4000 size used in our original simulations. Simulations were generated based on the implementation from the PopSim consortium – an open-source population genetics project (StdpopSim; Adrion et al. 2020). In addition to models used in our original submission (Hsieh et al. 2019), we performed new simulations using the following published models for individual modern human population groups to explore possible biases due to misspecification of underlying demographic models (**Supplementary Table 9**). For each model, we generated 10,000 whole-genome replications and computed Tajima's *D* values using the same 2,000 bp windows.

Supplementary Table 9. Summary of additional models used for coalescent simulations.

Super population	Reference	Model ID in StdpopSim
African	Jouanous et al. 2017	OutOfAfrica_4J17
American	Browning et al. 2011	AmericanAdmixture_4B11
East Asian	Jouanous et al. 2017	OutOfAfrica_4J17
European	Jouanous et al. 2017	OutOfAfrica_4J17
Middle Easterner	Gladstein and Hammer 2019	AshkSub_7G19
Melanesian	Malaspinas et al. 2016	Melanesian_6H19
Siberian	Hsieh et al. 2017	-
South Asian	Terhorst et al. 2019	-
Archaic hominin	Jacobs et al. 2019	PapuansOutOfAfrica_10J19

To summarize the overall performance of these models, we consider comparisons between the observed and simulated genome-wide statistics. We first would like to note that the main model used for our original submission (Hsieh et al. 2019) did show a significant correlation between

observed and simulated data using Watterson's theta statistic (**Response to Review Figure 1; copied from Hsieh et al. 2019, Fig. S9**). However, we recognized that this model was inferred specifically for Africans, East Asians, and Melanesians and was derived from different population samples than those used in the current study.

Response to Review Figure 1. (Copied from Hsieh et al. 2019, Figure S9) Whole-genome simulations accurately capture the local genetic diversity in the real data. Estimates of nucleotide diversity (Watterson's theta) were computed using windows of 100 SNVs for both the real and simulated data. Pearson's correlation coefficient test was used to assess the agreement between real and simulated whole-genome data.

Overall, the simulated Tajima's *D* distributions based on different coalescent models qualitatively recapitulate the observed data (**Supplementary Fig. 32**); however, in all cases we note that there are clear differences across all populations. These differences are likely, at least in part, because each of our super-population groups includes multiple geographically closed populations in the Human Genome Diversity Project, and, also in part, because these models are inferred using different population samples in individual studies (e.g., the 1000 Genomes Project and Simons Genome Diversity Project).

Supplementary Fig. 32. Comparisons of genome-wide Tajima's D statistic values between observed and simulated data using models listed in Supplementary Table 9. Tajima's D values are computed using the same 2,000 bp windows across the genome as described in the main text.

To determine if our original observations of selection at the *TCAF* locus still hold under these new models, we compared the observed Tajima's D values to those computed from these new simulations. Consistent with our original findings, in both archaic groups, we observed significantly negative Tajima's D values from the data compared with those from the simulations (**Supplementary Fig. 33**). Within modern humans, no populations show significantly negative Tajima's D values compared with simulated data; however, we note that the American population samples do have significantly larger Tajima's D values than those based on simulations (**Supplementary Fig. 33**). In addition, we notice that inconsistent with our original finding, the *TCAF* Tajima's D values of Melanesians becomes insignificant after accounting for multiple testing.

Supplementary Fig. 33. Comparisons of Tajima's D statistic values between observed and simulated $TCAF$ data using models listed in Supplementary Table 9. Tajima's D values were computed using 2,000 bp windows as described in the main text. P values were computed by testing if the observed values are significantly negative than those from simulated data, except for the American and Melanesian groups, where their p values represent if the observed Tajima's D values are significantly positive (italic) than those from the simulated data. Bold fonts indicate Bonferroni's p values < 0.05.

In summary, application of these different models recapitulates two of our original observations. We find that $TCAF$ Tajima's D values in the two archaic and Native American groups are 1) significantly different from those of the genome, and 2) significantly negative and positive, respectively, compared to their simulated $TCAF$ values. Thus, our new data provide additional evidence for and are consistent with our original findings of opposing selection forces in archaic and modern hominins.

Given these new results, we now revise the RESULTS and DISCUSSION sections of our manuscript as follows.

(RESULTS) Lines 290-313: "To formally test whether natural selection plays a role in these unusual patterns in both modern and archaic humans, we performed a selection test using the Tajima's D statistic and coalescent simulations under plausible demographic models to assess significance (**Methods**). In several human groups, we noticed that the Tajima's D values at

TCAF significantly deviate from those across the whole genome (**Fig. 5c**). Specifically, the Tajima's *D* values of the archaic hominins are significantly negative compared to the rest of the genome (median Tajima's *D* < -0.99; Bonferroni's *p* values $\leq 4.15 \times 10^{-7}$), regardless of the presence of the Denisovan individual in the test. In contrast, while the *TCAF* Tajima's *D* distributions among most modern human groups are compatible with those of the genome, there are two notable exceptions: Native American (median Tajima's *D* = 1.62; Bonferroni's *p* = 5.1×10^{-5}) and Melanesian (median Tajima's *D* = 0.69, Bonferroni's *p* = 0.01406) samples (**Fig. 5c**).

To account for biases in the Tajima's *D* values due to demographic effects, we simulated data using multiple population-specific models published recently (**Supplementary Table 9; Methods**) and computed Tajima's *D* values across 2,000 bp windows. Overall, the simulated Tajima's *D* distributions qualitatively recapitulate the observed data; however, in all cases we note differences in individual populations (**Supplementary Fig. 32**). These differences are likely due, in part, to the presence of multiple geographically closed populations in each supergroup, and, also in part, due to different population samples underlying the demographic models used in our study. Consistent with the results of the comparison between the *TCAF* locus and the whole genome, in both archaic hominin groups, we observe significantly negative Tajima's *D* values from the data compared with those from the simulations (Bonferroni's *p* values $\leq 6.62 \times 10^{-6}$; **Supplementary Fig. 33**). Within modern humans, no populations show significantly negative Tajima's *D* values compared with simulated data though we do note that the Native American samples show significantly more positive Tajima's *D* values than those based on simulations (Bonferroni's *p* value = 3.32×10^{-11} ; **Supplementary Fig. 33**), possibly as a result of balancing selection at this locus in this population."

(DISCUSSION) Lines 433-442: "In contrast, Neanderthal and Denisovan show a paucity of genetic variation, and while the sample size is still limited, this observation is unlikely to change with the sequencing of additional archaic genomes. The strongly negative Tajima's *D* and Fay and Wu's *H* values at this locus compared with those of the genome-wide and simulated values, which accounts for differences explained by demographic history, suggest that the observation of the single-haplotype without duplication in the archaic samples is highly unlikely under neutrality. We note that our selection inference is limited by our understanding regarding the demography of these archaic hominins and the underlying models and should be further tested in the future by incorporating new archaic samples and demographic models. Nevertheless, given our inference results, we argue that positive selection has reduced genetic diversity at the *TCAF* locus in these archaic hominin lineages."

It would also be good to add an explanation for why negative Tajima's *D* values are interpreted as positive selection rather than strong purifying selection.

As suggested by the reviewer, to further investigate if the negative Tajima's *D* values are more likely to be driven by positive selection, rather than purifying selection resulting in a reduction of genetic diversity among the archaic groups, we applied Fay and Wu's *H* test (Fay and Wu 2000) to the same genomic regions examined by Tajima's *D* test. Briefly, unlike the Tajima's *D* test where only folded allele frequency distribution is used to test selection, Fay and Wu's *H* statistic was specifically designed to identify an excess of high-frequency derived alleles at the locus of

interest – a signature of positive selection but not purifying selection. It, thus, requires outgroup information to utilize the full unfolded allele frequency distribution. In general, a negative H value indicates an excess of high-frequency derived alleles and is more consistent with positive selection without the confounder of a folded allele frequency distribution.

We computed the H statistic values using chimpanzee (Kronenberg et al. 2018) as the outgroup. We note that every simulated H distribution is centered around the zero (as expected), but its distribution is heavily left-skewed (long tails at the negative side; **Supplementary Fig. 34**), suggesting the relative conservativity of this test. While both the Neanderthal and archaic hominin groups are qualitatively negative, only the archaic hominin group is significant compared to the simulated distribution (Bonferroni's p value = 0.00917). This difference is likely due to a lack of power for this test, given that there are only three samples in the Neanderthal group. Importantly, when jointly considering Tajima's D and Fay and Wu's H values at $TCAF$, we find that both the Neanderthal and archaic hominin groups have distinct patterns between the observed and simulated data (**Supplementary Fig. 35**). The joint density of these two statistics from the observed $TCAF$ data largely falls within the negative range, which is different from the simulated $TCAF$ data, and are, thus, consistent with a signature of positive selection.

Supplementary Fig. 34. Comparisons of Fay and Wu's H values between observed and simulated $TCAF$ data using models listed in Supplementary Table 9. Fay and Wu's H values were computed using 2,000 bp windows as described in the main text. P values were

computed by testing if the observed values are significantly negative (green) than those from simulated data. Bold fonts show Bonferroni's p values < 0.05 . Note that only the Neanderthal and archaic hominin groups (blue IDs) have significantly negative Tajima's D values (Supplementary Fig. 33).

Supplementary Fig. 35a. Two-dimensional joint density plot showing distinct patterns of Tajima's D and Fay and Wu's H values between observed and simulated TCAF data for the Neanderthal group ($n=3$). Simulations were generated using the model from Jacob et al. (2019) listed in Supplementary Table 9.

Supplementary Fig. 35b. Two-dimensional joint density plot showing distinct patterns of Tajima's D and Fay and Wu's H values between observed and simulated TCAF data for

the archaic hominin group (n=4). Simulations were generated using the model from Jacob et al. (2019) listed in **Supplementary Table 9**.

In summary, while our analyses are still likely affected by uncertainty in the true demographic history of the archaic groups examined here, we find that the observed Tajima's D values at the *TCAF* locus are significantly negative compared with those of the whole genome (**Fig. 5C**) and simulated values (**Supplementary Fig. 33**). Furthermore, the observations of negative Fay and Wu's H values in the archaic groups provide additional support for positive selection, rather than purifying selection, being the cause of a reduction of genetic diversity at the *TCAF* locus (**Supplementary Figs. 34-35**). These new results further strengthen our conclusion on positive selection at the *TCAF* locus in the archaic hominins.

We now revise our RESULTS and DISCUSSION sections as follows.

(RESULTS) Lines 314-328: "To further investigate if the observed negative Tajima's D values in the archaic hominins are more likely to be driven by positive selection, rather than purifying selection resulting in a reduction of genetic diversity, we computed the Fay and Wu's H statistic ³⁰. Briefly, a negative H value indicates an excess of high-frequency derived alleles at the locus of interest—a signature of positive selection but not purifying selection (**Methods**). While both the Neanderthal and archaic hominin groups are qualitatively negative ($H \leq -0.86$), only the archaic hominin group is significant compared to the simulated distribution (Bonferroni's p value = 0.00917; **Supplementary Fig. 34**). The smaller sample size of the Neanderthal group ($n=3$) is likely limiting the power of the H statistic. In addition, we note that while the simulated H distribution under neutrality is centered around the zero as expected, its distribution is heavily left-skewed (**Supplementary Fig. 34**), suggesting that this test is relatively conservative. Of note, for both the Neanderthal and archaic hominin groups, the joint density of these two statistics from the observed *TCAF* data largely falls within the negative range, which is distinctively different from that of simulated data (**Supplementary Fig. 35**) and consistent with a signature of positive selection and unlikely explained by the demographic history of these hominins. Taken together, our results suggest significantly distinct signals of selection in both humans and archaic hominins."

(DISCUSSION) Lines 435-438: "The strongly negative Tajima's D and Fay and Wu's H values at this locus compared with those of the genome-wide and simulated values, which accounts for differences explained by demographic history, suggest that the observation of the single-haplotype without duplication in the archaic samples is highly unlikely under neutrality."

References:

- 1) Hsieh P, Vollger MR, Dang V, Porubsky D, Baker C, Cantsilieris S, Hoekzema K, Lewis AP, Munson KM, Sorensen M, Kronenberg ZN. Adaptive archaic introgression of copy number variants and the discovery of previously unknown human genes. *Science*. 2019 Oct 18;366(6463).

- 2) Adrion JR, Cole CB, Dukler N, Galloway JG, Gladstein AL, Gower G, Kyriazis CC, Ragsdale AP, Tsambos G, Baumdicker F, Carlson J. A community-maintained standard library of population genetic models. *Elife*. 2020 Jun 23;9:e54967.
- 3) Jouganous J, Long W, Ragsdale AP, Gravel S. Inferring the joint demographic history of multiple populations: beyond the diffusion approximation. *Genetics*. 2017 Jul 1;206(3):1549-67.
- 4) Browning SR, Browning BL, Daviglus ML, Durazo-Arvizu RA, Schneiderman N, Kaplan RC, Laurie CC. Ancestry-specific recent effective population size in the Americas. *PLoS genetics*. 2018 May 24;14(5):e1007385.
- 5) Gladstein AL, Hammer MF. Substructured population growth in the ashkenazi jews inferred with approximate bayesian computation. *Molecular biology and evolution*. 2019 Jun 1;36(6):1162-71.
- 6) Malaspinas AS, Westaway MC, Muller C, Sousa VC, Lao O, Alves I, Bergström A, Athanasiadis G, Cheng JY, Crawford JE, Heupink TH. A genomic history of Aboriginal Australia. *Nature*. 2016 Oct;538(7624):207-14.
- 7) Hsieh P, Hallmark B, Watkins J, Karafet TM, Osipova LP, Gutenkunst RN, Hammer MF. Exome sequencing provides evidence of polygenic adaptation to a fat-rich animal diet in indigenous Siberian populations. *Molecular biology and evolution*. 2017 Nov 1;34(11):2913-26.
- 8) Terhorst J, Kamm JA, Song YS. Robust and scalable inference of population history from hundreds of unphased whole genomes. *Nature genetics*. 2017 Feb;49(2):303-9.
- 9) Jacobs GS, Hudjashov G, Saag L, Kusuma P, Darusallam CC, Lawson DJ, Mondal M, Pagan L, Ricaut FX, Stoneking M, Metspalu M. Multiple deeply divergent Denisovan ancestries in Papuans. *Cell*. 2019 May 2;177(4):1010-21.

The evidence of selection in humans is also weak. Tajima's D Pvalues are based on simulations. Figure 5 legend and Methods state that the simulations are based on 1,000 different demographic models. Why 1,000 models? What models, since the reference is to one? How well do they recapitulate the Tajima's D values in 2,000bp segments in the genome? Several of the populations with significant Pvalues have complex demography so this is important. Another way to strengthen the evidence would be to show that the structure of adjacent 2,000bp segments have Tajima's D values that agree with expectations under natural selection.

We would like to clarify that we generated simulations based on 1,000 sets of different demographic parameters drawn in the multivariate covariance-variation matrix from Hsieh et al. (2019) for the Old World populations. However, the reviewer correctly points out that several populations in the current study, such as the Siberians and Native Americans, indeed have a more complex demography.

In our revision we now include simulations based on several recently published (and more realistic) models for these populations **Supplementary Table 9**. In particular, the American demographic model specifically accounts for recent admixture events among Africans, Europeans, and Native Americans (Browning et al. 2011), and the Siberian model (Hsieh et al. 2017) was inferred using the same population (the Yakut) used in the current study. With these new simulations, as stated earlier we found that the genome-wide Tajima's *D* distributions between the data and simulations are qualitatively compatible but have noticeable differences (**Supplementary Fig. 32**). Importantly, we find only the Native American group shows significantly positive values compared to the simulated distribution at the *TCAF* locus.

Table S9. Summary of additional models used for coalescent simulations.

Super population	Reference	Model ID in Stdpopsim
African	Jouganous et al. 2017	OutOfAfrica_4J17
American	Browning et al. 2011	AmericanAdmixture_4B11
East Asian	Jouganous et al. 2017	OutOfAfrica_4J17
European	Jouganous et al. 2017	OutOfAfrica_4J17
Middle Easterner	Gladstein and Hammer 2019	AshkSub_7G19
Melanesian	Malaspinas et al. 2016	Melanesian_6H19
Siberian	Hsieh et al. 2017	-
South Asian	Terhorst et al. 2019	-
Archaic hominin	Jacobs et al. 2019	PapuansOutOfAfrica_10J19

With these new results, we have revised our RESULTS, DISCUSSION, and METHODS sections in our manuscript accordingly as follows.

(RESULTS) Lines 301-313: “To account for biases in the Tajima's *D* values due to demographic effects, we simulated data using multiple population-specific models published recently (**Supplementary Table 9; Methods**) and computed Tajima's *D* values across 2,000 bp windows. Overall, the simulated Tajima's *D* distributions qualitatively recapitulate the observed data; however, in all cases we note differences in individual populations (**Supplementary Fig. 32**). These differences are likely due, in part, to the presence of multiple geographically closed populations in each supergroup, and, also in part, due to different population samples underlying the demographic models used in our study. Consistent with the results of the comparison between the *TCAF* locus and the whole genome, in both archaic hominin groups, we observe significantly negative Tajima's *D* values from the data compared with those from the simulations (Bonferroni's p values $\leq 6.62 \times 10^{-6}$; **Supplementary Fig. 33**). Within modern

humans, no populations show significantly negative Tajima's D values compared with simulated data though we do note that the Native American samples show significantly more positive Tajima's D values than those based on simulations (Bonferroni's p value = 3.32×10^{-11} ; **Supplementary Fig. 33**), possibly as a result of balancing selection at this locus in this population.

(DISCUSSION) Lines 438-442: "We note that our selection inference is limited by our understanding regarding the demography of these archaic hominins and the underlying models and should be further tested in the future by incorporating new archaic samples and demographic models. Nevertheless, given our inference results, we argue that positive selection has reduced genetic diversity at the *TCAF* locus in these archaic hominin lineages."

(METHODS) Lines 623-630: "To test for signals of natural selection, we computed the Tajima's D ⁶² and Fay and Wu's H statistics³⁰ over 2,000 bp windows over the 52.3 kbp unique diploid sequences for individual populations as well as the archaic samples. Because these statistics are sensitive to population history, we performed a large coalescent simulation using both MaCS⁶³ and msprime (v0.7.4)⁶⁴ and demographic models as described previously^{25,65} that were implemented in the Python package Stdpopsim (v0.1.2)⁶⁶. For each of the models, we generate 10,000 whole-genome simulations to account for past histories between modern and archaic hominins and among multiple African and non-African populations (**Table S9**) and use chimpanzee²⁷ as the outgroup."

References:

- 1) Hsieh P, Vollger MR, Dang V, Porubsky D, Baker C, Cantsilieris S, Hoekzema K, Lewis AP, Munson KM, Sorensen M, Kronenberg ZN. Adaptive archaic introgression of copy number variants and the discovery of previously unknown human genes. *Science*. 2019 Oct 18;366(6463).
- 2) Browning SR, Browning BL, Daviglus ML, Durazo-Arvizu RA, Schneiderman N, Kaplan RC, Laurie CC. Ancestry-specific recent effective population size in the Americas. *PLoS genetics*. 2018 May 24;14(5):e1007385.
- 3) Hsieh P, Hallmark B, Watkins J, Karafet TM, Osipova LP, Gutenkunst RN, Hammer MF. Exome sequencing provides evidence of polygenic adaptation to a fat-rich animal diet in indigenous Siberian populations. *Molecular biology and evolution*. 2017 Nov 1;34(11):2913-26.

The results for functional effects of changes in the region are interesting but also not definite. The functional relevance of the eQTLs is ambiguous. If the functional relevance of *TCAF1* and *TCAF2* is by regulating TRPM8, are the tissues where eQTLs are identified relevant in terms of TRPM8? Does TRPM8 express at high levels in these tissues? Likewise, the associations with disease are only nominal and in single populations. More evidence is needed to show that they are biologically relevant.

The reviewer raised an interesting suggestion regarding the expression patterns between *TCAFs* and *TRPM8*. Using data from both GTEx (release v8) and the Human Protein Atlas (HPA) database (<https://www.proteinatlas.org>), *TCAF2* and *TRPM8* are both highly expressed in liver and prostate tissues (**Supplementary Fig. 37**) but not particularly expressed in *TCAF*-related tissues (thyroid, lung, nerve-tibial, etc.) that we highlighted in the paper. Interestingly, both liver and prostate tissues are where haplogroup H5 has significant and positive effect sizes on the *TCAF2* expression relative to haplogroup H4 (**Figure 6C**). Using the single-cell RNA-seq data from the HPA database, we further examined the *TCAF2* and *TRPM8* expression levels in different cell types in liver and prostate tissues (**Supplementary Figs. 38-39**). Both *TCAF2* and *TRPM8* are almost exclusively expressed in clusters (clusters c-0, 2, 5, 7 and 11) corresponding to hepatocytes of liver and in prostate glandular (clusters c-6 and 13) and endothelial cells (cluster c-7). These findings together imply that the difference in *TCAF2* expression between the H4 and H5 haplogroups could play a role in differentially regulating *TRPM8* in the liver and prostate tissues among human groups.

We agree with the reviewer that while these observations on the association between *TCAF* haplotype-tagging SNVs and diseases are interesting, they are far from definitive and it is obvious that more work will need to be done before more definitive statements regarding disease associations can be made. Thus, in our revision, we remove all relevant association results in this regard and revise the RESULTS and DISCUSSION section as follows.

(RESULTS) Lines 349-362: “Given the common presence of these two super haplogroups in humans (**Fig. 6a, b**), we collected multi-tissue eQTL data from the GTEx Project (release v8) and found that all four tagging SNVs are *TCAF2* eQTLs (**Fig. 6c**). Among tissues, the H5 haplogroup (alternative) alleles exclusively increase the *TCAF2* expression in liver and prostate tissues (p values $< 1 \times 10^{-5}$), while the H4 haplogroup (reference) alleles are significantly associated with increasing *TCAF2* (p values $< 1 \times 10^{-10}$) expression in tissues, such as thyroid and tibial nerve tissues, in addition to others (**Fig. 6c**). In addition, using the GTEx data and the HPA database, we find that both *TCAF2* and *TRPM8* are highly expressed in liver and prostate (**Supplementary Fig. 37**). To further examine the expression of *TCAF2* and *TRPM8* across different cell types in liver and prostate tissues, we used the single-cell RNA-seq data from the HPA (**Supplementary Figs. 38-39**). Both *TCAF2* and *TRPM8* are almost exclusively expressed in clusters (clusters c-0, 2, 5, 7 and 11) corresponding to hepatocytes of liver and in prostate glandular (clusters c-6 and 13) and endothelial cells (cluster c-7). These findings together imply that the difference in *TCAF2* expression between the H4 and H5 haplogroups may play a role in differentially regulating *TRPM8* in the liver and prostate tissues among human groups.”

(DISCUSSION) Lines 443-464: “The opposing effects on the *TCAF2* expression found between H4 and H5 haplogroups and the observations of high expression of both *TCAF2* and *TRPM8* in liver and prostate tissues are particularly intriguing. While the classical function of *TRPM8* is as cold and menthol receptors for sensory neurons, it was first identified as an overexpressed gene in prostate cancer ^{42,43} and is known for its role in controlling calcium flux and homeostasis—an important physiological process in cold signaling ⁴⁴. Moreover, recent studies demonstrated the role of *TRPM8* in energy homeostasis; specifically, it is required for precise thermoregulation in response to cold in the fed and fasting states in mice ⁴⁵. Not only is *TRPM8* required for cold

detection in mice ⁴⁵, but *TRPM8*-knockout mice raised at mild cold temperatures develop late-onset obesity and display a metabolic dysfunction with a higher content of lipid droplets in liver ⁴⁶. This implies that the TRPM8 ion channel, in addition to its role in detecting environmental cold, is a known regulator of insulin homeostasis through neuronal control of liver insulin clearance and, thus, required for precise thermoregulatory responses to cold and fasting ^{47,48}. Interestingly, a rare variant within the *TCAF* locus (chr7:143735931) was recently reported to be associated with type 2 diabetes in sub-Saharan African populations ⁴⁹. Given this variant's restricted geographic distribution and low frequency in the GTEx data (allele frequency: ~0.4%; GTEx portal as of 2020.07.21), it explains neither the observed patterns of diversity nor the selection and the eQTL signals observed here. Therefore, if the *TCAF* haplotype and structural diversity affect an individual's ability to regulate TRPM8 ion-channel gating and, thus, body temperature, we hypothesize that the antagonistic forces of selection promote adaptations to a cold environment or an enhanced ability to dynamically adapt to changing conditions, such as diet. Further experimental investigations on the adaptive significance and role of the different *TCAF* copies in regulating *TRPM8* in humans are required. These are now possible given that the structure and diversity of this dynamic locus has been more fully resolved. ”

Supplementary Fig. 37. Normalized gene expression for *TCAF2* and *TRPM8* across 55 tissues and 6 blood types using the consensus dataset in the Human Protein Atlas database (<https://www.proteinatlas.org>).

Fig. 6c. Differential effect sizes on the *TCAF2* gene expression between the haplogroups H4 and H5. H5 has negative effects on the *TCAF2* expression in thyroid, nerve-tibial, blood but has positive effects in prostate and liver.

Supplementary Fig. 38. RNA expression of *TCAF2* (left) and *TRPM8* (right) in the single cell type clusters identified in the liver tissue visualized by a UMAP plot (top) and a bar chart (bottom) using the Human Protein Atlas database (<https://www.proteinatlas.org/>).

Top: UMAP plot visualizes the cells in each cluster (c-0 to c-16); where each dot corresponds to a cell. Color is based on cell type groups used in the Cell Type Atlas, and color intensity represents the individual cells according to % of max

$(\log_2(\text{read_count}+1)/\log_2(\max(\text{read_count})+1)*100)$ in five different bins

(<1%, <25%, <50%, <75%, ≥75%). Bottom: The bar chart shows normalized RNA expression in each cell type cluster. Color-coding follows those on the top of the plot (i.e., cell type). Black and grey outlines of the bars indicate the *TCAF2* and *TRPM8* expression, respectively.

Supplementary Fig. 39. RNA expression of *TCAF2* (left) and *TRPM8* (right) in the single cell type clusters identified in the prostate tissue visualized by a UMAP plot (top) and a bar chart (bottom) using the Human Protein Atlas database

(<https://www.proteinatlas.org/>). Top: UMAP plot visualizes the cells in each cluster (c-0 to c-16); where each dot corresponds to a cell. Color is based on cell type groups used in the Cell Type Atlas, and color intensity represents the individual cells according to % of max ($\log_2(\text{read_count}+1)/\log_2(\text{max}(\text{read_count})+1)*100$) in five different bins (<1%, <25%, <50%, <75%, $\geq 75\%$). Bottom: The bar chart shows normalized RNA expression in each cell type cluster. Color-coding follows those on the top of the plot (i.e., cell type). Black and grey outlines of the bars indicate the *TCAF2* and *TRPM8* expression, respectively.

The correlation of CN with latitude is interesting. Exploring this in more detail would strengthen the paper because the authors propose temperature and diet as selection forces. The authors could for example ask if the correlation is higher than observed in other segments of the genome.

We explored this more in detail by performing a random sampling of 1,000 SD loci and saw that ~22% of these loci also significantly correlate with latitude and ~19% of them have correlation coefficients > 0.23 (**Supplementary Fig. 18**). Interestingly, no loci have p values for the correlation tests as extreme as that of the acceptor site where we identified interlocus gene conversion (IGC) (p value = $2.54e-11$; **Supplementary Fig. 17**). Furthermore, only seven sampled loci have correlation coefficients > 0.23 and remain significant after Bonferroni's correction (p value = 0.007; **Supplementary Fig. 19**). This suggests the observed correlation between the paralog-specific copy number (CN) at the putative IGC loci and the latitudinal locations of human populations is highly unlikely under a null expectation and suggestive of potential selection.

Supplementary Fig. 18. Tests of correlation between copy number estimates and latitudinal locations of populations for 1,000 random segmental duplication loci. Left panel: the distribution of Pearson's correlation coefficients for the randomly drawn loci, and the vertical line indicates the observed Pearson's correlation coefficient of 0.23 for the putative IGC acceptor locus (**Supplementary Fig. 17**). Right panel: the distribution of p values for this

random set, where the dashed and solid vertical line indicates the observed p value at the IGC locus and the 0.05 cutoff, respectively.

Supplementary Fig. 17. Significant correlation between paralog-specific *TCAF* CN at the putative IGC locus and latitudinal locations of human populations. The plot shows a significant positive correlation between the paralog-specific CN estimates at the IGC acceptor site and latitudinal locations of 54 human populations. The latitudes of these 54 populations can be found in **Supplementary Table 8** in the supplementary materials of the original submission.

Supplementary Fig. 19. Correlation between paralog-specific *TCAF* CN at the putative IGC locus and latitudinal locations of human populations. Data were taken from **Figure S18** and replotted with the absolute correlation coefficients and Bonferroni's p values (n=1000) for the random sampled loci. The seven loci that show absolute coefficients > 0.23 (the observed Pearson's correlation coefficient of 0.23 for the putative IGC acceptor locus in

Supplementary Fig. 17) and with corrected p values < 0.05 are highlighted in red and are the most extreme. Note that none of the seven outliers have p values as small as the one observed in **Supplementary Fig. 17** (Bonferroni's p value = $2.54e-8$).

We added this new analysis to the supplementary material and revised the text as follows to highlight these new observations.

(RESULTS) Lines 218-225: “While the observed correlations between latitudes of individual populations and paralog-specific copy numbers at the IGC sites may be the result of natural selection, such strong correlations are relatively common among similar loci across the genome (p value = 0.22; 1,000 random SD loci of genomic shuffling) (**Supplementary Fig. 18**), suggesting demographic history among populations also contributes to this observation. However, only seven of the 1,000 sampled SD loci have correlation coefficients > 0.23 and remain significant after Bonferroni's correction (**Supplementary Fig. 19**), indicating that such a correlation is unlikely under a random sampling (p value = 0.007).”

Minor points.

The color scheme in Figure 1C could be better. It is hard to see differences among few copy numbers with small figures and similar colors.

We revised Fig. 1c to improve the clearness as suggested by the reviewer.

Figure 1. Copy number variation of *TCAF* SDs in a collection of diverse human and nonhuman samples. Copy numbers were estimated using Illumina read-depth based genotyping method. (A) Copy number (CN) heat maps, where each row represents the CN of a sample over the *TCAF* locus. The colored arrows (A, B, and C) represent the three major SD blocks in this region. The white area in the middle represents the gap present in the human reference genome (GRCh38). The two white boxes show a putative interlocus gene conversion event that correlates with latitudinal locations of populations (Figure S16-S18). Dark green bars at the bottom indicate the unique diploid sequences used for downstream phylogenetic and population genetic analyses (chr7:143,501,000-143,521,000, chr7:143,729,525-143,741,525, chr7:143,875,000-143,895,000 [GRCh38]). (B) Distributions of the overall *TCAF* CN genotypes among samples from nonhuman great apes, archaic hominins, and modern humans using a representative region (chr7:143,615,002–143,624,482). (C) Geographic distribution of the overall *TCAF* CN genotypes in the 54 Human Genome Diversity Project (HGDP) populations and the nonhuman great ape samples. Geo-coordinates for the populations are listed in **Supplementary Table 8**. Pie charts show the CN frequency distribution for a given population. Note that the color scheme is slightly different from those in the CN heat maps in Fig. 1a.

The estimate for the age of DupB is older than estimates of the age of a single copy region. Is this meaningful or informative? Is that single copy region representative of our expectations in the genome? Why in p.244 the reference for this single copy locus is to Figure 2?

- 1) Genetic variation among paralogs records the history among the duplicate copies, while variants in the unique single-copy region are allelic variation representing demographic histories among populations and/or species. Therefore, the elder age of the DupB indicates the initiation time of *TCAF* SD expansion, and the younger age of the single-copy region represents the lower bound of the complete formation of *TCAF* haplotype diversity in hominins.
- 2) At Line 244, the reference for the single-copy locus should be Fig. 1a, not to Fig. 2. We thank the reviewer for pointing out this error and now revise the sentence as follows.

(RESULTS) Lines 242-244: “While a few differences in topology were noted among the inferred phylogenies (**Supplementary Figs. 21-23**), these estimates, with the exception of DupB, overlapped coalescent estimates obtained from a 12.3 kbp single-copy unique region (**Fig. 1a, the middle green segment at the bottom**), which”

The authors would help the reader if they explained haplotype-based PCA. In Figure 5A the main figure and the zoom figure look different. Why is that?

- 1) We now add a description of our haplotype-based PCA in the METHODS section.
(METHODS) Lines 632-636: “To perform a haplotype-based PCA, we first recode haplotype bases as numeric codes, where 0 and 1 (and 2 if needed) are ancestral (using chimpanzee as the outgroup) and derived alleles, respectively. We performed the haplotype-based PCA following the suggestion described in Browning et al 2016 ⁶⁸ but using the R package irlba (<https://cran.r-project.org/web/packages/irlba/index.html>) to calculate principal components.”
- 2) The zoomed-in portion of **Fig. 5a** was rotated and jittered to highlight the eight archaic haplotypes. We now revise the figure as follows without these manual modifications.

Figure 5. Archaic hominin versus human haplotype diversity. Haplotypes were inferred using 1,275 SNVs in the three unique diploid sequences around the TCAF SD region (Figs. 1 and 2). (A) Haplotype-based principal component analysis was performed, followed by haplotype clustering and cluster visualization using the dimension-reduction technique, t-SNE (Methods). On the t-SNE plot, each dot/triangle is a haplotype and colored according to population/species origin. Neanderthal and Denisovan haplotypes are indicated by the black and blue triangles, respectively. Numbers and ellipses in the 3D t-SNE plots indicate individual clusters (see also **Supplementary Figs. 24-29**). The zoom-in above the 3D t-SNE shows that all archaic haplotypes are in close proximity to each other and associate with cluster 1. (B) The maximum likelihood phylogeny was constructed using 10 randomly selected haplotypes from the 12 inferred clusters, in addition to eight archaic and one chimpanzee haplotypes. Note that the branch length of chimpanzee (dashed line) is truncated by 90% of its actual length for the purpose of illustration. (C) Comparisons of the Tajima's *D* distributions between the TCAF locus and the entire genome in human and archaic populations. Tajima's *D* statistics are computed for individual archaic and modern groups across the entire genome (grey) versus TCAF (red) based on 2 kbp windows. P values are empirically determined by testing if the observed values at TCAF are significantly negative (green) than those from the genome, except for the American and Melanesian group, where their p value represents if the observed *D* values at TCAF are significantly positive (*italic*) than the those from the genome. Bold fonts indicate Bonferroni's p values < 0.05.

Reference:

- 1) Browning SR, Grinde K, Plantinga A, Gogarten SM, Stilp AM, Kaplan RC, Avilés-Santa ML, Browning BL, Laurie CC. Local ancestry inference in a large US-based Hispanic/Latino study: Hispanic community health study/study of Latinos (HCHS/SOL). *G3: Genes, Genomes, Genetics*. 2016 Jun 1;6(6):1525-34.

L 382. Is the importance of TCAF in thermal regulation and sensing established? If it is, explaining how that works would strengthen the paper and make the interpretation less dependent on TRPM8 and its expression.

Unfortunately, the role of thermal regulation of *TCAFs* was only established recently based on the JCB study (Gkika, D. et al. 2015) that shows the trafficking of the cold sensor *TRPM8* is regulated by TCAF1/2.

As suggested by the reviewer, in our revision, we revise our DISCUSSION as follows.

(DISCUSSION) Lines 462-464: “Further experimental investigations on the adaptive significance and role of the different *TCAF* copies in regulating *TRPM8* in humans are required. These are now possible given that the structure and diversity of this dynamic locus has been more fully resolved. ”

Reference:

- 1) Gkika, D. et al. TRP channel-associated factors are a novel protein family that regulates TRPM8 trafficking and activity. *J Cell Biol* 208, 89-107 (2015).

REVIEWER COMMENTS

Reviewer #1 (Remarks to the Author):

I'd like to thank the author for their absolutely convincing and exceptionally thorough response to my comments.

I have no further questions or comments. But I would like to point out the following semi-related paper about a cold-adaptation allele that was found in Inuits and originated in Denisovans that the authors may find of interest:

Racimo et al. MBE (2017) "Archaic adaptive introgression in TBX15/WARS2"

I was not involved in this work and do not think it needs to be cited.

Reviewer #2 (Remarks to the Author):

Hsieh have revised their manuscript taking into account both reviews. I focus on the response to my comments, even if some of their analyses are adequately used in both rebuttal letters. Their responses to some of my comments have improved the manuscript. The work performed to address my major comments is adequate, and I appreciate their efforts. Unfortunately the results provided do not support the conclusions of signatures of positive selection in Neanderthals and the claim of opposing selective signatures in Neanderthals and humans.

The authors performed additional simulations and an empirical analysis to show that this locus has been under positive selection in Neanderthals. I have some concerns:

- The main evidence of positive selection is that the Tajima's D values in the TCAF locus are "significantly negative when compared with the rest of the genome" and "significantly negative than those from the genome". This is shown by comparing the distribution of Tajima's D in the locus with the distribution in

the simulations and the genome. The problem is that being “more negative” than simulations or that the genome is by no means evidence of positive selection. Only being in the 1% or 5% tail of the distribution in simulations or the genome is considered evidence of positive selection. This is the standard in the field, and for a good reason. Many factors can increase or reduce Tajima’s D in the absence of positive selection. Only strong departures from the “neutral distribution” are difficult to explain under neutrality and require positive selection as an explanation. The authors do not show that this is the case in this locus, and in fact Supplementary Figure 30 and Supplementary Figure 33 suggest that this is not the case. In the absence of windows in the tail of the “neutral” distribution, I see no evidence of positive selection.

- Conversely, the evidence for opposing signatures in humans is based on Melanesians and Native Americans showing “significantly positive D values”. But Supplementary Figure 30 shows that these distributions are similar to the genome ones although somewhat biased. As mentioned, moderate bias is not significant evidence of selection, only being in the 1% or 5% of the distribution is.

- To distinguish positive from purifying selection the authors calculated Fay and Wu’s H. This is a good approach, and they use simulations to test its departure from neutral expectations. Unfortunately, the evidence of positive selection in Neanderthals is even weaker with H than with Tajima’s D. The H for Neanderthals in the TCAF locus is not different from the H in simulations. There is no signature of selection. The H for all archaic hominids is “significantly negative” but this is likely due to Denisovans bringing their own alleles, distorting the site frequency spectrum and generating negative H. As I noted before these genomes do not represent a population so comparisons with simulations need to be careful. The rebuttal letter does not include a comparison with the genome and in the absence of this information we cannot discard this possibility. Supplementary Figures 35a and 35b show again a distribution of H in TCAF that falls within the simulated distribution although slightly biased. As with Tajima’s D, only H that departs from expectations under neutrality can be considered evidence of positive selection.

To summarize there is no evidence in the data presented for the sentence “The strongly negative Tajima’s D and Fay and Wu’s H values at this locus compared with those of the genome-wide and simulated values”.

The suggestive correlation with latitude remains interesting. The fact that 22% of the loci show correlation too shows that demography and co-ancestry influence the analysis. But this locus seems unusual and its signatures are somehow strong. If the authors controlled the phylogenetic signature in the data this could be a nice result.

Responses to Reviewer Comments

REVIEWER COMMENTS

Reviewer 1:

I am not certain why the authors do not provide a Supplementary Figure similar to Figure 5c for Fay and Wu's H (if I have not overlooked it), which feels a more direct way to address one of referees #2's points. I support adding such a Figure or a table with empirical p-values should referee #2 deem them crucial. I feel that the claims are well supported but that it is not unreasonable to be interested in empirical p-values, especially when those were already produced for Tajimas D in the paper.

Reviewer 2:

Hsieh et al., have re-submitted a response and a modified manuscript. I thank the authors for considering my points and performing some new analyses. Unfortunately, the letter and revision do not address my main concern. I do not comment on the advantages and disadvantages of using the genomic distribution or using simulations. I make no claim in support of an outlier approach. In fact, I refer to a null model based either on the genome or on simulations. To repeat "The problem is that being "more negative" than simulations or that the genome is no evidence of positive selection." This is not a test of polygenic adaptation. It is a test of neutrality in a single locus. A skew from the mean is not considered evidence of selection.

Regardless of whether the null (neutral) distribution is based on the genome or simulations, only a value of the statistic that falls in the tail of the neutral distribution allows us to reject the null and provides evidence for positive selection. A value of the statistic that falls within the neutral distribution conforms with neutral expectations and does not provide evidence of natural selection. This is true even if the statistic is skewed with respect to the mean because for single loci a skew from the mean still conforms with expectations under neutrality. After all, much of a distribution is more negative than its mean. I agree that outlier empirical methods have problems but if we interpret being in the 85% of the null distribution as a signature of positive selection then we interpret that 15% of the genome is under positive selection. A genomic window being surrounded by other similarly skewed windows is not surprising under neutrality due the effects of linkage in genealogies. Thus the distribution of values in a locus being somehow skewed is not surprising. Evidence of selection would come from some of these windows having values that are unexpected under neutrality.

The issue applies to positive selection in Archaic and to balancing selection in America. In both cases the authors interpret a skew from the mean as "departures from the genome-wide empirical distributions and the expectation of neutrality under realistic demographic models". In my view the claim for "opposing selective forces" that remains in the title is not supported by the data.

The paper includes an impressive amount of work on a complex genomic region that may be functionally important. That is exceptional work as far as I can tell. I also appreciate the authors taking the time to study the evolution of the locus. I wish I could be more positive but unfortunately I do not think that the claim for opposing selection is supported by the data.

And cross-commenting by Reviewer 1:

“The request of reviewer #2 is completely reasonable and should take seconds for the authors to deliver. S/he would like to see that windows in the *TCAF* locus are in the extreme tail of either simulated or observed distribution of Tajima's *D* or Fay and Wu's *H* genome-wide. Eyeballing the plot in Fig 5c, this does seem to be the case, at least for Tajima's *D* compared to the observed genome-wide values. The authors just need to put these numbers and those for the other test and the other, simulated null in a table and adjust how they phrase their discussion if the *TCAF* windows should fail to end up in the tail of these distributions.”

We thank Referee #2's clarification for the concern in question and also for the suggestions from Referee #1. Based on the provided suggestions, we provide a Figure similar to **Figure 5c** for Fay and Wu's *H* (**Supplementary Fig. 35**). In addition, we also provide tables showing numbers of windows in the *TCAF* locus that are in the extreme tail of either simulated or observed distribution of Tajima's *D* or Fay and Wu's *H* (**Supplementary Table 10**).

As suggested by Referee #1, the distributions for the *TCAF* locus are significantly different and map to the extreme tail of either simulated or empirical genome-wide distribution of Tajima's *D* or Fay and Wu's *H* statistic. Consistent with our other observations, the values of Fay and Wu's *H* in the *TCAF* locus are significantly more negative both in the archaic hominin and Neanderthal groups (**Supplementary Fig. 35**), a classic signature of positive selection. Moreover, we observe that multiple *TCAF* windows fall within the extreme tails of both the Tajima's *D* and Fay and Wu's *H* distributions for both observed and simulated cases (**Supplementary Table 10**). In contrast, the Native American results map once again to the tails of the simulated and observed distributions but, instead, show significant positive Tajima's *D* values consistent with the signature for balancing selection. As Referee #2 pointed out, because these windows having values that are unexpected under neutrality, they provide further evidence for selection.

Together, our results are in agreement with the expectations of selection from both the referees that multiple *TCAF* windows are indeed in the extreme tail of either simulated or observed genome-wide distribution of the two statistics. As we noted in our Discussion section, although there are other explanations which we acknowledge in the discussion, we believe these observations are consistent with opposing forces of positive and balancing selection in archaic hominin and at least one modern human population and respectfully request that the title not be changed. We have modified the Results section to include these new findings in the main text and the figures/tables in the Supplementary.

(RESULTS) Lines 332-334: Consistent with the Tajima's D results, we observe significantly more negative Fay and Wu's H values from the *TCAF* locus than those from the genome-wide data in both the archaic hominin and Neanderthal groups (**Supplementary Fig. 35**).

(RESULTS) Lines 347-354: Not only are the distributions of the two statistics for the *TCAF* locus significantly different from the neutral expectations for archaic hominins, we also find that multiple *TCAF* windows fall in the extreme (left/negative) tails of both the Tajima's D and Fay and Wu's H genome-wide distributions for both observed and simulated data (**Supplementary Table 10**), providing evidence for positive selection. In contrast, the Native American results map once again to the (right/positive) tails of the observed and simulated distributions but, instead, show significant positive Tajima's D values consistent with the signature for balancing selection (**Supplementary Table 10**). Because these windows having values that are unexpected under neutrality, they provide further evidence for selection.

Supplementary Fig. 35. Comparisons of Fay and Wu's H values between observed *TCAF* and genome-wide data. Fay and Wu's H values were computed using 2000 bp windows as described in the main text. Bonferroni's p values were computed by testing if the observed values are significantly more negative (green) than those from simulated data. Bold fonts show Bonferroni's p values < 0.05 . Note that only the Neanderthal and archaic hominin groups (blue IDs) have significantly negative Tajima's D values (**Supplementary Fig. 34**).

Table S10. Numbers of windows in the *TCAF* locus in the extreme tails of genome-wide distribution of observed and simulated Tajima's *D* and Fay and Wu's *H*. Data were directly extracted from **Figure 5c** and **Supplementary Figs. 34-35**. A total of 29 windows in the *TCAF* locus were used in the comparisons. The numbers in parentheses are the thresholds used to determine if the values of the statistics for individual windows fall within the extreme tails.

Archaic hominin	Tajima's D		Fay and Wu's H	
	#windows at the 1% tail	#windows at the 5% tail	#windows at the 1% tail	#windows at the 5% tail
Whole genome	5/29 (≤ -1.51)	10/29 (≤ -1.31)	3/29 (≤ -4.07)	8/29 (≤ -2.36)
Simulated genome	0/29 (≤ -2.04)	1/29 (≤ -1.68)	2/29 (≤ -6.74)	3/29 (≤ -3.71)
Simulated TCAF locus	1/29 (≤ -1.61)	11/29 (≤ -1.21)	1/29 (≤ -7.04)	3/29 (≤ -3.86)

Neanderthal	Tajima's D		Fay and Wu's H	
	#windows at the 1% tail	#windows at the 5% tail	#windows at the 1% tail	#windows at the 5% tail
Whole genome	9/29 (≤ -1.29)	12/29 (≤ -1.13)	4/29 (≤ -2.93)	11/29 (≤ -1.33)
Simulated genome	0/29 (≤ -1.96)	1/29 (≤ -1.49)	0/29 (≤ -8.27)	2/29 (≤ -4.53)
Simulated TCAF locus	0/29 (≤ -1.45)	3/29 (≤ -1.36)	0/29 (≤ -7.15)	2/29 (≤ -4.53)

Native American	Tajima's D	
	#windows at the 1% tail	#windows at the 5% tail
Whole genome	0/29 (≥ 3.00)	3/29 (≥ 2.26)
Simulated genome	9/29 (≥ 1.98)	17/29 (≥ 1.41)
Simulated TCAF locus	9/29 (≥ 1.93)	16/29 (≥ 1.47)

REVIEWERS' COMMENTS

Reviewer #3 (Remarks to the Author):

The authors have done an impressive amount of work on the manuscript and revision. I was not an original reviewer of this manuscript, but after careful examination of both reviewers' comments and the authors' revisions, I feel that the reviewers' concerns have been adequately addressed. The conclusion for positive selection is justified by the new data presented, and I have no other concerns about the manuscript.

Responses to Reviewer Comments

REVIEWERS' COMMENTS

Reviewer #3 (Remarks to the Author):

The authors have done an impressive amount of work on the manuscript and revision. I was not an original reviewer of this manuscript, but after careful examination of both reviewers' comments and the authors' revisions, I feel that the reviewers' concerns have been adequately addressed. The conclusion for positive selection is justified by the new data presented, and I have no other concerns about the manuscript.

We appreciate all three reviewers for their help to improve our manuscript.